# Optimization design of centrifugal impeller based on Bezier surface and FFD space grid parameterization

Yesong Wang[1], Zixuan Sun[1], Jisheng Liu[2], Manxian Liu[2], Yong Zhou[3]*

1 School of Mechanical Engineering, Jiangsu University of Science and Technology, Zhenjiang, China,
2 School of Mechanical Engineering, University of Science and Technology Beijing, Beijing, China, 3 China Industrial Control Systems Cyber Emergency Response Team, Beijing, China

* gxzhouyong@126.com

**Data Availability Statement:** All relevant data are within the manuscript.

**Funding:** The author(s) received no specific funding for this work.

## Abstract

To enhance the aerodynamic performance of centrifugal impellers, this study presents an advanced optimization design methodology. This methodology addresses the challenges associated with numerous design variables, inflexible configurations, and low optimization efficiency. We propose two distinct spline function parameterization techniques: a global mapping model for Bezier surfaces and a local mapping model for Free-Form Deformation (FFD) control bodies. We investigate the impact of these parameterization methods on blade geometry configuration and aerodynamic performance. By integrating these two parameterization approaches with multi-objective evolutionary algorithms and Computational Fluid Dynamics (CFD) techniques, we enable global and local optimization of centrifugal compressor blades. The optimization results demonstrate a 1.77% enhancement in isentropic efficiency under rated operating conditions, a 7.8% increase in surge margin, a 1.6% improvement in isentropic efficiency under normal operating conditions, and an 11.8% enhancement in surge margin. Through two optimization stages, the optimization space for blade geometry is thoroughly explored, enhancing solution quality and contributing to the advancement of impeller mechanical optimization design theory.

## Introduction

Parametric methods are the primary focus of research in the field of three-dimensional aerodynamic optimization design of impeller machinery [1–4]. Traditional parametric methods primarily concentrate on the two-dimensional profile lines of the blades. Painter, R et al. [5] investigated the effect of PAT performance by modifying the shape of the impeller tip leading edge curve and increasing the impeller inlet width, etc. The experimental study found that the optimal impeller leading edge modification significantly increased the PAT efficiency by 3–3.5%, while widening of the inner shroud reduced the efficiency by 1.2%.

Zhang et al. [6]optimised the efficiency of a centrifugal pump by distributing spline curves with multiple points on the hub and shroud surfaces, controlling the vane thickness and vane angle and combining them with an optimisation algorithm to increase the hydraulic efficiency

**Competing interests:** The authors have declared that no competing interests exist.

by 4.19%. Xu [7] utilized Bezier curves to parametrically represent the trailing edge of a centrifugal compressor blade. The optimization results demonstrated that enhancing the trailing edge bending angle effectively improved the isentropic efficiency and total pressure ratio while reducing trailing losses.

While the traditional parametric method using profile lines is widely employed, it suffers from limitations such as a limited number of profiles for parametric expression. As a result, it often leads to insufficient optimization space, lack of flexibility, and an inability to fully optimize the overall geometric shape of the blades. Increasing the number of profile lines would exponentially increase the design space and could easily lead to dimensionality issues. Furthermore, the traditional design method lacks radial constraints, which hinders the generation of smooth blade shapes. This approach no longer satisfies the requirements for achieving refined designs of high-performance impellers.

To address these challenges and achieve precise blade modifications, this paper proposes a two-stage optimized design method for centrifugal compressor blades based on spline functions. The proposed method leverages the advantages of surface parameterization and spatial mesh parameterization to establish both the Bezier surface model [8–10] and the free-form deformation (FFD) model [11–13]. Additionally, the study investigates the influence of Bernstein basis and B-spline basis on blade reshaping. By employing two stages of global optimization and local optimization, the method achieves flexible and refined reshaping of centrifugal compressor blades with a reduced number of global fast optimizationdesign variables. This approach significantly improves the comprehensive performance of centrifugal impellers.

## Parameterization of centrifugal compressor blades using spline basis functions

This paper introduces two novel parametric methods, Bezier profile parameterization and FFD 3D mesh parameterization, for optimizing the design of complex surface blades. These methods enhance efficiency and flexibility in blade shaping design by employing high-dimensional parametric techniques.

### Selection of the basis function

Bernstein basis functions and B-spline basis functions [14–17] are commonly used for spline functions. Bernstein basis functions exhibit global characteristics, meaning that any change in a control point's position affects the entire spline curve, as seen in the blue curve of Fig 1. Even minor adjustments to control points result in significant reshaping. B-spline basis functions, on the other hand, have strong local support, meaning changes are localized.

In contrast, B-spline basis functions exhibit strong local support. A point on a k-order B-spline curve is influenced by a maximum of k control points, independent of others. Moving a control vertex, like Pi, affects only a localized portion of the curve within its corresponding interval, resulting in minimal overall reshaping. Fig 1 illustrates this with the red curve.

Bernstein and B-spline basis functions share properties like geometric invariance and higher-order smooth continuity, but their unique strengths make them ideal for different applications. Bernstein basis functions excel in global blade reshaping due to their ability to affect the entire curve, while B-spline basis functions provide localized control for precise adjustments.

### Bernstein-based Bezier surface global parameterization method

Fig 2 illustrates the Bernstein-based Bezier surface parameterization process, which involves the following steps: (l) Obtain the leading and trailing edge points of the blades using surface

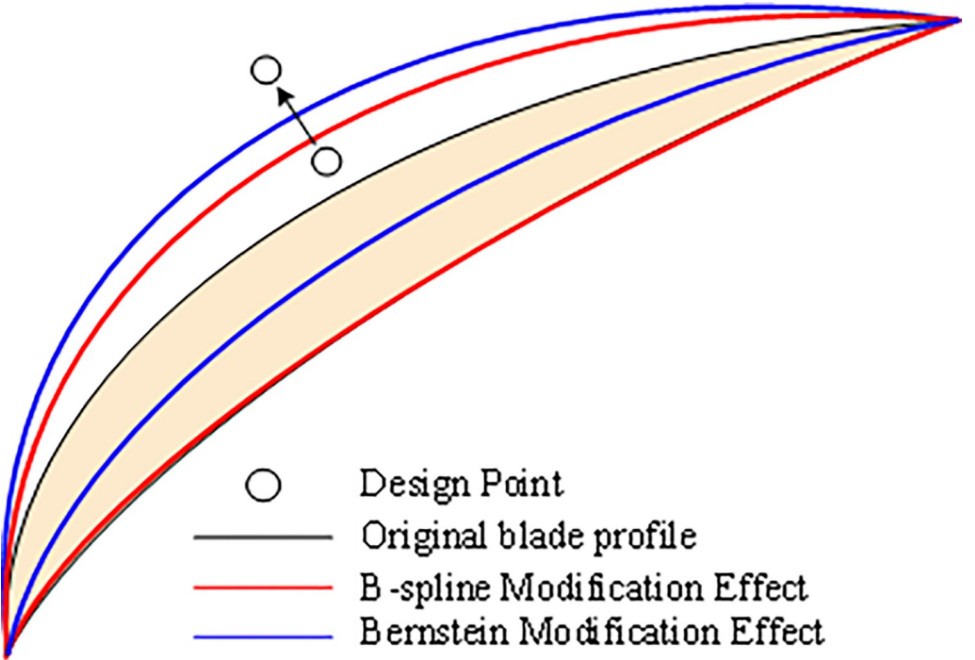

**Fig 1. Effect of Bernstein and B-spline on profile control.**

data from the suction and pressure surfaces. (II) Encrypt the data points for each cross-sectional blade type through transverse interpolation. (III) Normalize the chord lengths of each section of the original blade. Parameterize the points of each cross section on the original suction and pressure surfaces using the chord lengths, as shown in Eqs (1) and (2).

$$\xi_{i,j} = \frac{\sum_{c=1}^{i} l_c}{L_j} \tag{1}$$

$$\eta_{i,j} = \frac{\sum_{r=1}^{j} l_r}{L_i} \tag{2}$$

a7 is the length of the j-th chord in the c-th radial section. Lj represents the sum of the chord lengths in the j-th section. a8 denotes the length of the r-th segment of the chord in the i-th section along the axial direction, and Li is the sum of the chord lengths of the segments in the i-th section. Where $\xi_{i,j}$ and $\eta_{i,j}$ represent the normalized horizontal and vertical coordinates of the chord length, respectively, $i \in (1, N_p)$, $N_p$ refers to the number of points in each radial section. While $j \in (1, N_s)$ and $N_s$ refers to the total number of radial sections. $l_c$ is the length of the $j$-th chord of the $c$-th section in the radial direction. $L_j$ represents the sum of the chord lengths of the $j$-th section. $l_r$ denotes the length of the $r$-th segment of the chord on the $i$-th section in the axial direction, and $L_i$ is the sum of the chord lengths of the segments on the $i$-th section. (IV) Generate two unitized Bezier surfaces defined by Eqs (3)–(7):

$$\bar{S} = \sum_{k=0}^{n} \left\{ \sum_{l=0}^{m} P_{k,l} N_l^m(v) \right\} N_k^n(u) \tag{3}$$

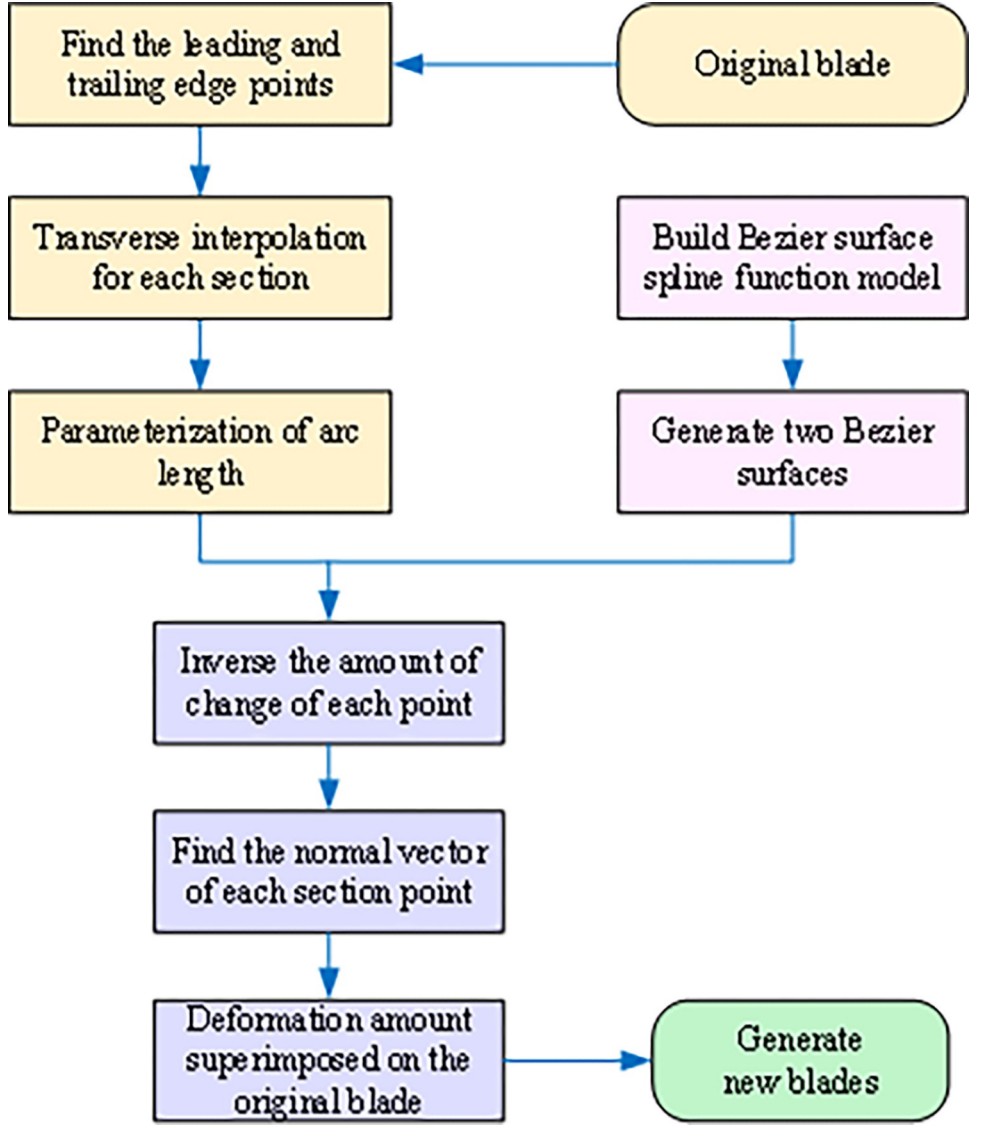

**Fig 2. Effect of Bernstein and B-spline on profile control.**

$$N_l^m(v) = C_l^m v^l (1-v)^{m-l} \tag{4}$$

$$N_k^n(u) = C_k^n u^k (1-u)^{n-k} \tag{5}$$

$$C_l^m = \begin{cases} \dfrac{m!}{(m-l)!\,l!} & \text{if } 0 \leq l \leq m \\ 0 & \text{if not} \end{cases} \tag{6}$$

$$C_k^n = \begin{cases} \dfrac{n!}{(n-k)!k!} & if\ 0 \le k \le n \\ 0 & if\ not \end{cases} \qquad (7)$$

Where $\bar{S}$ is the coordinate of each point on the Bezier surface ($\bar{S} = (S_x, S_y, S_z)$, $S_x = \xi_{i,j}$, $S_y = \eta_{i,j}$, $S_z = \delta$); $P_{k,l}$ is the Bezier surface control vertex, and the total number of control points is ($m$ +1)+($n$+1), $N_l^m(v)$ and $N_k^n(u)$ are Bernstein basis functions, where $v$ and $u$ are two independent variables varying in the range [0,1]. $C_l^m$ is calculated from Eq (6) and $C_k^n$ is obtained from Eq (7).

(V) Set the optimization variables and define their ranges. Then, using the optimization algorithm, assign values to the control vertices of the two Bezier surfaces. Calculate the amount of variation and unit normal vectors for the original blade suction and pressure surface data points. (VI) Superimpose the changes in the suction and pressure surface data points onto the normal direction of the original surface. This results in new surface data points for both the suction and pressure surfaces, as shown in Eq (8).

$$y_{new} = y_{old} + \Delta s \qquad (8)$$

Where $y_{new}$ refers to the new blade coordinates, $y_{old}$ represents the original blade coordinates, and $\Delta s$ represents the directional shift from the Bezier surface corresponding to that point.

Bernstein-based Bezier surface parameterization significantly reduces the dimensionality of 3D blade design, enabling flexible modification of the entire blade with a minimal number of design points. This method uses fewer than twenty design variables, a dramatic improvement compared to traditional parameterization methods that require hundreds of variables.

This method's core principle is to treat the pressure and suction surfaces of the original blade as independent entities, allowing for individual modification. Unitized Bezier surfaces are then directly overlaid onto these surfaces. As illustrated in Fig 3, blade shape control is achieved by perturbing the Bezier surface. When the Bezier surface is flat (i.e., the perturbation amount is zero), the resulting blade generation surface replicates the original blade surface.

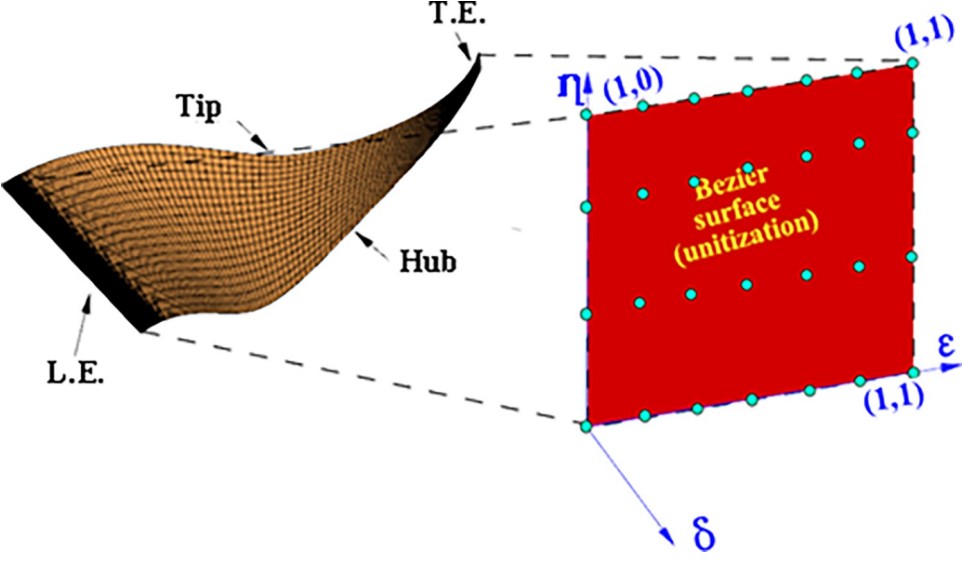

**Fig 3. The mapping relationship between blade surfaces and Bezier surfaces.**

By manipulating the optimization variables in one dimension, we can achieve a new three-dimensional blade design. During the superposition process, the four vertices of the Bezier surface align with the corresponding vertices of the original blade surface. Additionally, the high-order continuity of each point on the Bezier surface ensures a smooth optimized surface, minimizing flow losses caused by blade irregularities.

Bernstein-based Bezier surface parameterization offers a powerful approach for blade design, providing key advantages such as:

(I) Efficient Global Optimization: This method enables efficient optimization of the first-stage blade design on a global scale, facilitating effective exploration and optimization of the overall blade configuration. (II) Independent variation of suction and pressure surfaces: The parameterization allows for independent modifications of the suction and pressure surfaces, providing greater control over the blade's aerodynamic characteristics and enabling targeted improvements. (III) Integrated Thickness and Strength Considerations: This method ensures that blade thickness and mechanical strength requirements are considered during optimization, promoting structural integrity and durability. (IV) Radial Constraints for Smoothness: Geometric control parameters are subject to radial constraints, reducing the number of design variables and promoting the generation of smooth blade surfaces, minimizing flow losses and enhancing overall performance.

In conclusion, Bernstein-based Bezier surface parameterization offers a compelling approach for blade design, leveraging several key advantages. Its ability to facilitate efficient global optimization, enable independent surface variations, consider blade thickness and mechanical strength, and promote smooth surfaces through radial constraints contributes significantly to the effectiveness and performance of the optimized blade design.

## B-spline based FFD local parameterization method

FFD is an advanced extension of the Bezier surface parametric method, offering increased flexibility and deformability for arbitrary geometric shapes. It provides greater universality and degrees of freedom for local geometric configurations. As illustrated in Fig 4, the core concept involves placing the object to be modified within a mesh control body composed of a control point array. By applying external forces to the control body, the enclosed geometry can deform elastically, mirroring the control body's movements. Deformation of the control body is achieved by displacing its vertices.

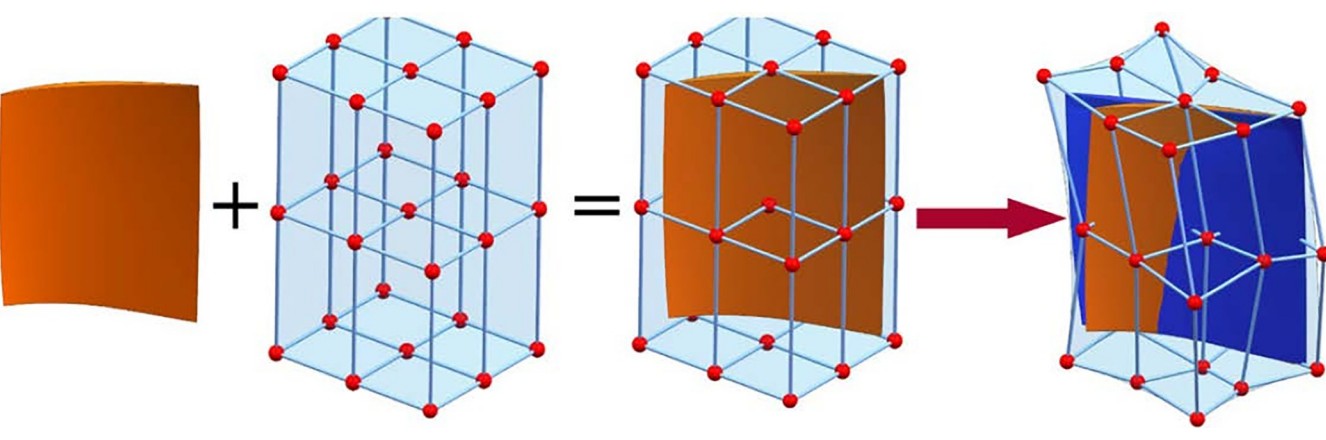

**Fig 4. FFD technology.**

Eq (9) defines the relationship between any point $\overrightarrow{X}$ on the surface and the control vertices of the B-spline basis function FFD control body mesh.

$$
\begin{cases}
\overrightarrow{X}(s,t,u) = \sum\limits_{i=0}^{l}\sum\limits_{j=0}^{m}\sum\limits_{k=0}^{n}\overrightarrow{P}_{i,j,k}B_{i,d}(s)B_{j,e}(t)B_{k,f}(u) \\
\quad s_{d-1} \le s \le s_{l+1} \\
\quad t_{e-1} \le t \le t_{m+1} \\
\quad u_{f-1} \le u \le u_{n+1}
\end{cases}
\tag{9}
$$

where $\overrightarrow{P}_{i,j,k}$ is the FFD control body mesh vertex. $(s,t,u)$ is the local coordinate of $\overrightarrow{X}$ within the control body. $l,m,n$ is the number of divisions of the FFD control frame in the three directions. $B_{i,d}(s)$, $B_{j,e}(t)$, $B_{k,f}(u)$ correspond to the B-spline basis functions of order $d,e,f$, respectively, which are derived by Boor-Cox, See Eqs (10)–(11).

$$
B_{i,1}(s) = \begin{cases} 1 & s_i \le s < s_{i+1} \\ 0 & Otherwise \end{cases}
\tag{10}
$$

$$
B_{i,d}(s) = \frac{s-s_i}{s_{i+d-1}-s_i}B_{i,d-1}(s) + \frac{s_{i+d}-s}{s_{i+d}-s_{i+1}}B_{i+1,d-1}(s)
\tag{11}
$$

The mathematical definition of $B_{j,e}(t)$ and $B_{k,f}(u)$ is similar to the principle of $B_{i,d}(s)$. The B-spline basis provides strong local support, meaning that moving a control vertex only affects the local geometry around that vertex, leaving other regions unchanged. This localized influence makes the B-spline basis FFD method ideal for local optimization, requiring only a small number of control vertices in the region of interest. This allows for precise manipulation of the blade geometry without altering the shape of unaffected areas. Additionally, local optimization can be performed without considering interface continuity, ensuring overall blade smoothness. This feature makes the B-spline basis FFD method well-suited for achieving flexible local deformation.

The modeling process for the centrifugal compressor blade using B-spline based FFD parameterization is illustrated in Fig 5. Initially, the entire blade is positioned within the FFD frame control body. The number of control vertices is then specified in the chordal, circumferential, and radial directions. To minimize the number of control vertices and narrow the search space, the control vertex layout should closely align with the object's geometric surface, as shown in Fig 6. Next, a B-spline basis function is selected, and a mapping model is established between the FFD control body vertices and the blade surface point cloud. This involves determining the local coordinates $(s,t,u)$ of each point on the surface relative to the control body. While the Newton iterative method is commonly used for solving local coordinates, its sensitivity to initial value selection and potential for local optima make it less robust. This paper proposes a robust and initial value-insensitive Monte Carlo algorithm to solve for local coordinates and establish an error model between the mapping function and the real blade data points, as described in Eq (12).

$$
Q(s,t,u) = \sum\limits_{i=0}^{l}\sum\limits_{j=0}^{m}\sum\limits_{k=0}^{n}\overrightarrow{P}_{i,j,k}B_{il}(s)B_{jm}(t)B_{kn}(u) - X_{real}
\tag{12}
$$

Initialize local coordinates $(s,t,u) = (s_0,t_0,u_0)$, calculation of $Q_0$. Choose a positive number $t$, generate a random vector $n$ on the interval a $[-t,t]$, calculation of

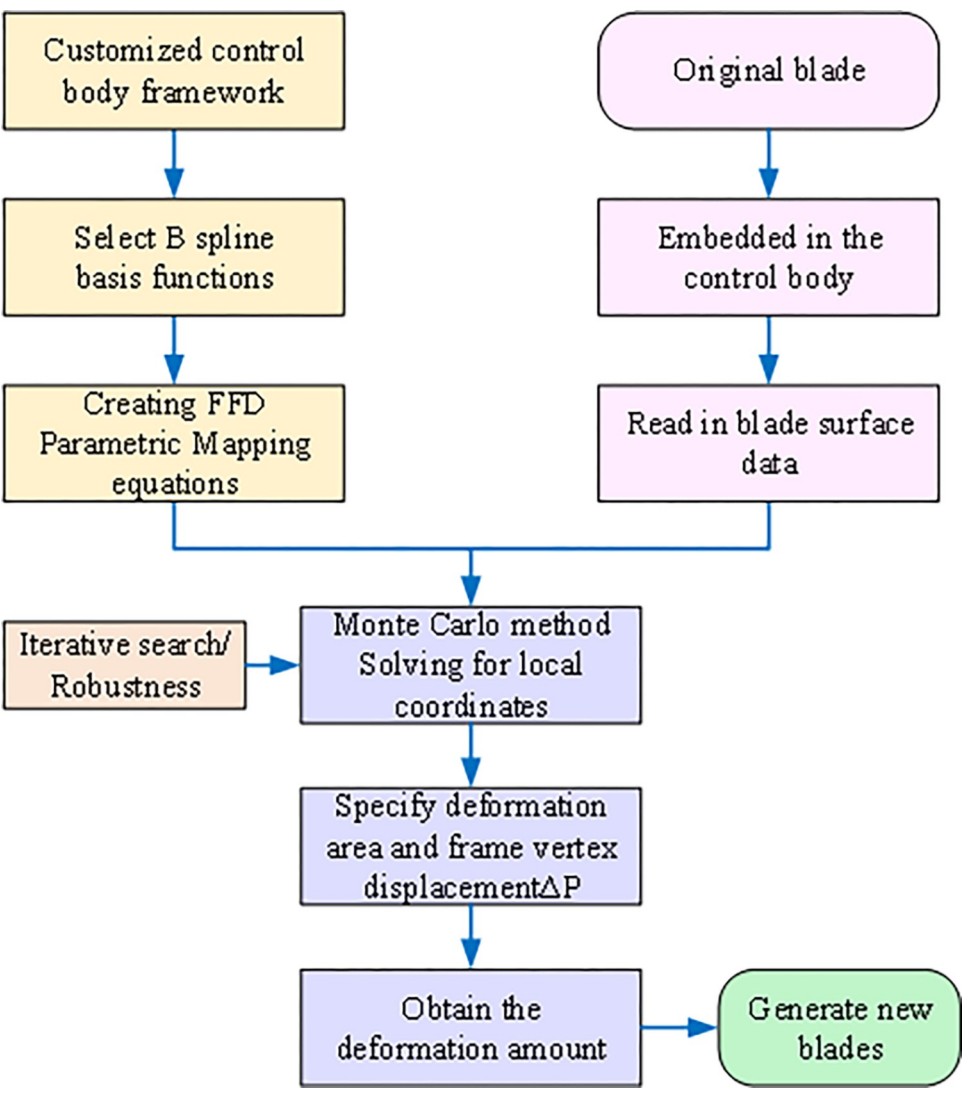

**Fig 5. FFD parametric method modeling flow chart.**

$Q_1 = Q_0(s_0 + n_s, t_0 + n_t, u_0 + n_u)$. When $Q_1 < Q_0$, let $(s, t, u) = (s_0 + n_s, t_0 + n_t, u_0 + n_u)$ and $Q_0 = Q_1$. If the randomly generated multiset random vector still does not satisfy $Q_1 < Q_0$, then let $t = t/2$; The cyclic operation stops when $Q_0 < \varepsilon$ is satisfied, at which point $(s,t,u) = (s_{best}, t_{best}, u_{best})$, and the correspondence between the surface and the local coordinates is obtained. $\vec{P}_{i,j,k}$ passes through $\Delta \vec{P}_{i,j,k}$ to obtain the new control vertex $\vec{P}\prime_{i,j,k}$. Using the consistency of local coordinates, the deformed object surface Cartesian coordinates $\overrightarrow{X}'$ are Eq (13).

$$\overrightarrow{X}'(s, t, u) = \sum_{i=0}^{l} \sum_{j=0}^{m} \sum_{k=0}^{n} \vec{P}\prime_{i,j,k} B_{il}(s) B_{jm}(t) B_{kn}(u) \tag{13}$$

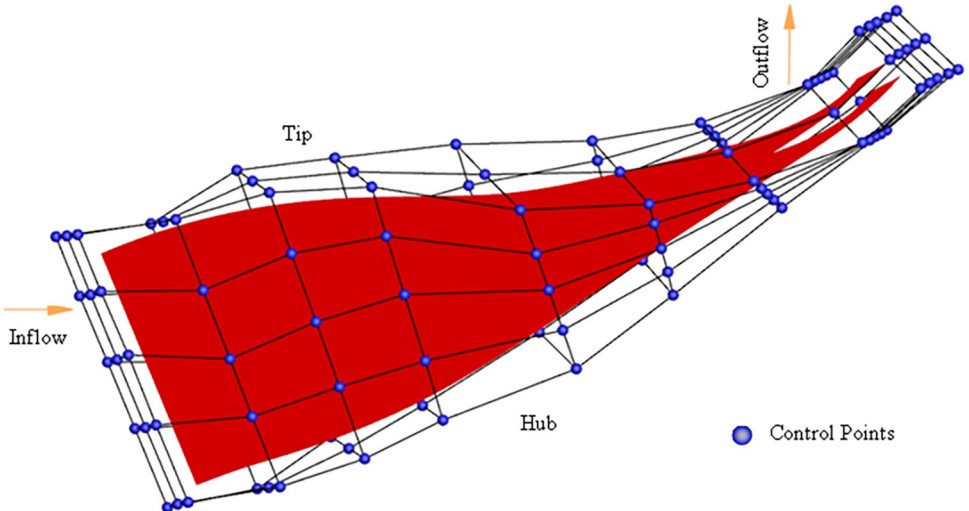

**Fig 6. Centrifugal compressor blade FFD method control body vertex distribution.**

## Centrifugal impeller multi-condition aerodynamic performance optimization study

### Optimization object

This study focuses on a centrifugal compressor impeller designed for a rated output of 100 kW and a peak speed of 100,000 rpm. The optimization targets the geometry of the main impeller blade and the splitting blade, while maintaining the shape of the end wall surface throughout the optimization process. The impeller model is shown in Fig 7, and Table 1 summarizes the aerodynamic and geometric parameters. **ROC** refers to the rated operating condition, while **NOC** represents the normal operating condition.

### Numerical calculation and verification

Numerical simulation techniques are widely used for investigating impeller machinery's internal flow. This study employs a one-equation turbulence model (the S-$A$ model) and computes temporal evolution using a fourth-order explicit Runge-Kutta scheme. A finite volume central difference scheme, incorporating second- and fourth-order artificial viscosity terms, is used to mitigate numerical oscillations during spatial discretization. To enhance convergence speed, the algorithm utilizes multiple meshes, local time steps, and hidden residuals. The mesh is generated using the Autogrid5 module in NUMECA [18–20], the main topology is H&I, the tip gap mesh is HO topology, the first layer of mesh near the wall is $1 \times 10^{-6}$ mm thick, $Y^+ \leq 5$. The total temperature at the inlet of the compressor is 293K, with a total pressure of 101325 Pa. The flow direction at the inlet is axial, and the outlet boundary condition is set as an average static pressure. By gradually increasing the back pressure, the calculation progresses from the surge point towards the near-stall point. The convergence point immediately preceding the first divergence point is identified as the near-stall point. A no-slip boundary condition is applied to the blade surface and end-wall.

To ensure grid quality during flow field calculations, grid independence was verified for a single channel at rated speed. The total number of grid points for two rows of blades, including the main blade and diverter blade, was varied: 300,000, 660,000, 1,030,000, and 1,500,000. Fig 8 shows the results, indicating that the isentropic efficiency error between calculations using

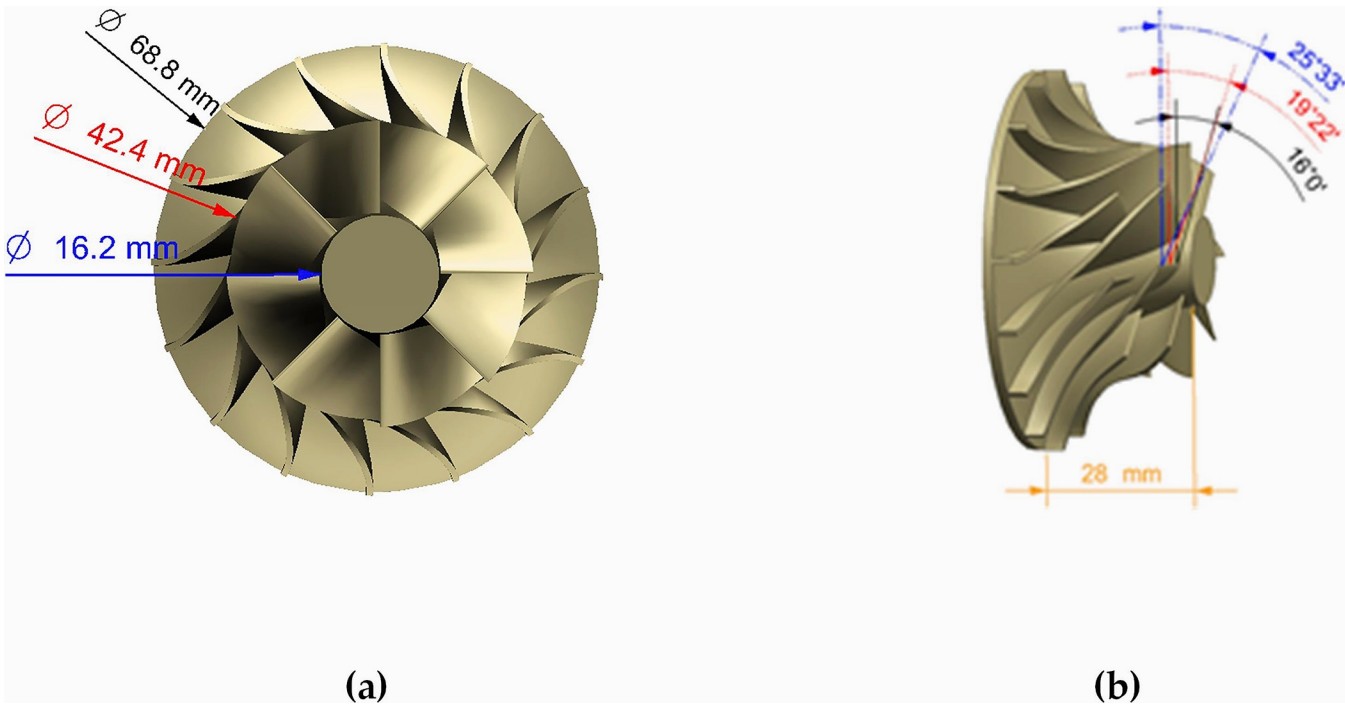

**(a)**                                                                            **(b)**

**Fig 7.** Original impeller geometry: (a) front view, (b) Side view.

1.03 million and 1.5 million grid points is only 0.02%. This suggests that once the total number of grid points reaches 1.03 million, further grid refinement has minimal impact on aerodynamic performance. To save time and cost, the subsequent optimization process utilized the 1.03 million grid structure as the grid template.

Numerical verification involved calibrating the computational method against experimental data from the Krain impeller [21]. Fig 9 compares the experimental data with CFD results, showing close agreement in isentropic efficiency. While a slight discrepancy exists in the total pressure ratio, this aligns with findings from Krain and other researchers [22]. This discrepancy arises because CFD calculations provide an average static pressure at the outlet, whereas experimental measurements yield a non-average static pressure. Consequently, the calculated pressure ratio is higher than the experimental value. Overall, the numerical simulation results accurately capture the trends of the performance curve. This method reliably analyzes changes in aerodynamic performance before and after impeller optimization, confirming the accuracy of the flow field calculations.

**Table 1. Aerodynamic parameters and geometrical dimensions.**

| Parameter | Value | Parameter | Value |
|---|---|---|---|
| Fluid medium | Air | Rated power/kW | 100 |
| Number of blades | 16(8+8) | Top clearance/mm | 0.3 |
| Radial outlet angle/˚ | 85 | Blade thickness /mm | 1 |
| Rpm @ ROC | 100,000 | Rpm @ NOC | 70,000 |
| Mass flow /(g/s) @ ROC | 118.32 | Mass flow /(g/s) @ NOC | 77.36 |
| Total Pressure Ratio @ ROC | 2.70 | Total Pressure Ratio @ NOC | 1.7 |
| Isentropic efficiency @ ROC | 83.54% | Isentropic efficiency @ NOC | 85.53% |

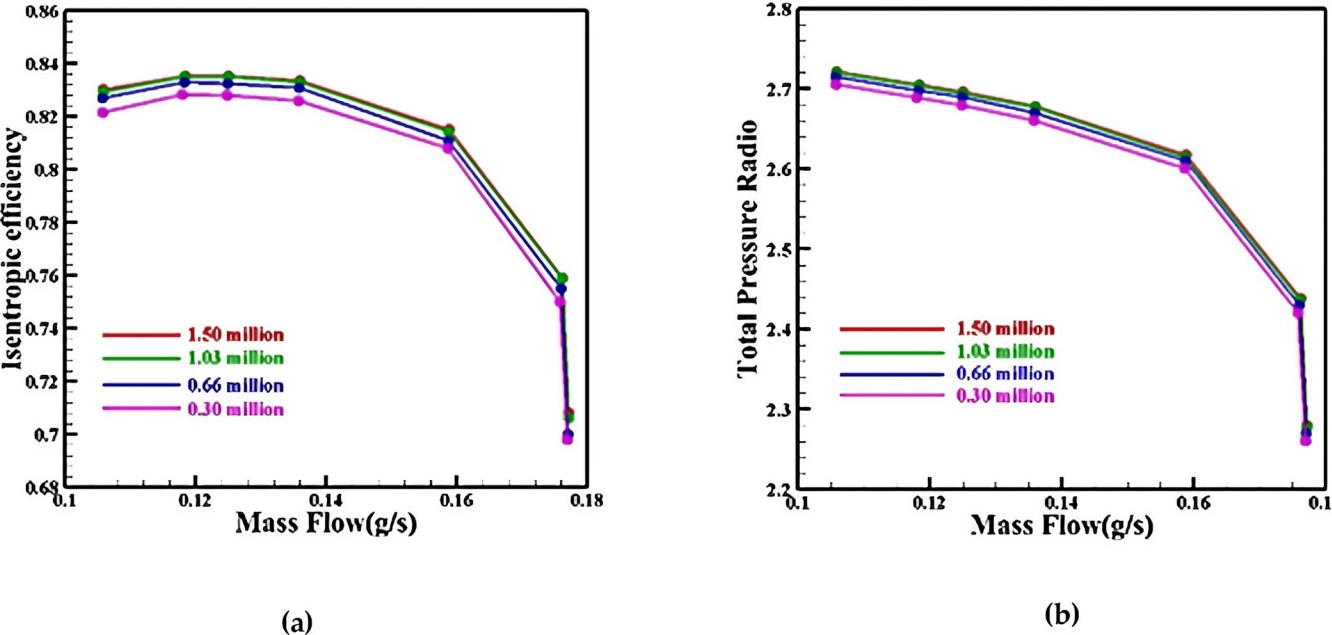

**(a)**

**(b)**

**Fig 8.** Grid-independence verification: (a) Mass Flow-Efficiency, (b) Mass Flow-Total Pressure Ratio.

## Optimization results and analysis

**Optimization strategy.** The optimization process, illustrated in Fig 10, comprises two distinct phases: global optimization and local optimization. In the first phase, global optimization using the Bernstein-based Bezier surface parameterization method rapidly identifies near-optimal solutions. The second phase focuses on refining specific areas with significant potential

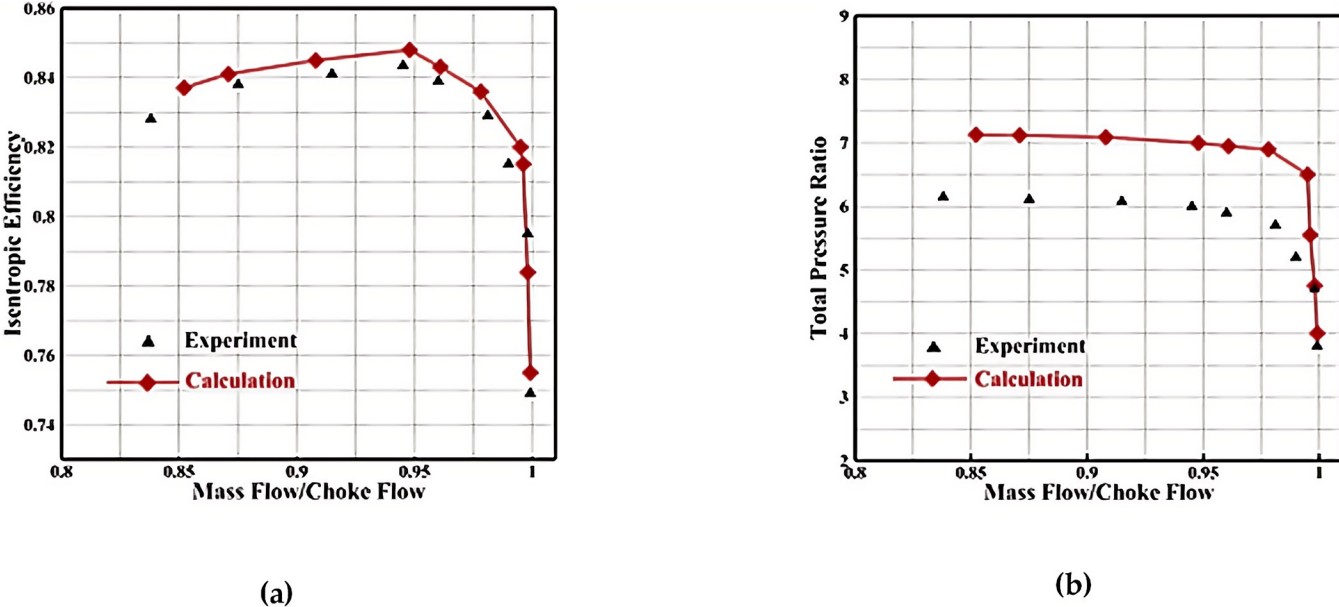

**(a)**

**(b)**

**Fig 9.** Comparison of numerical model calculation and experimental data: (a) Relative flow—Isentropic efficiency, (b) Relative flow—Total pressure ratio.

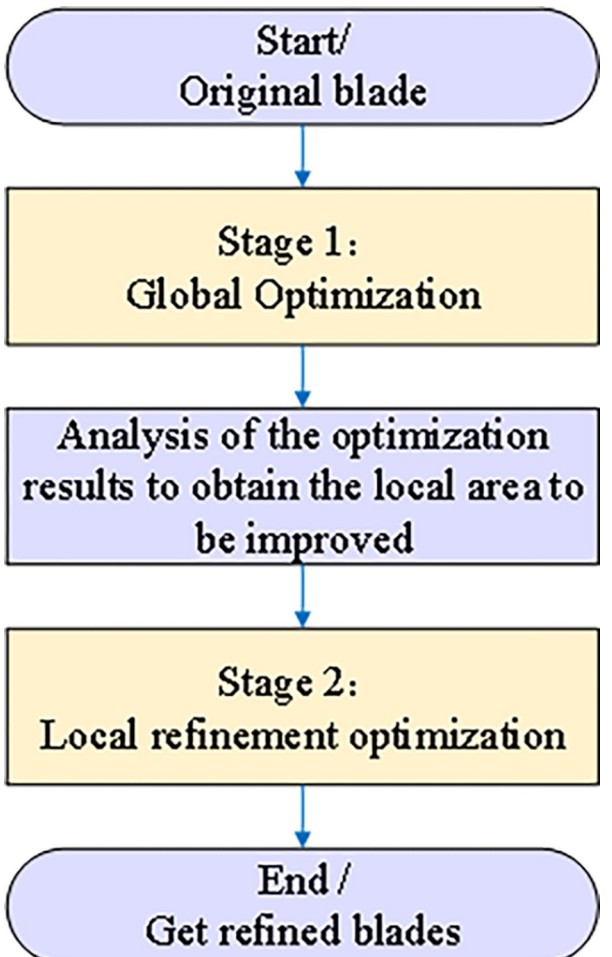

**Fig 10. Two-stage optimization process.**

for performance improvement, identified through analysis of the global optimization results. This local refinement utilizes the B-spline-based FFD parameterization method. By combining these two phases, the optimization process achieves a comprehensive enhancement of the blade's aerodynamic performance. The global phase narrows the search space to near-optimal solutions, while the local phase fine-tunes the blade's geometry in targeted areas to further enhance its performance.

**First-stage blade global optimization.** The NSGA-IV algorithm, a genetic algorithm, offers several advantages over traditional gradient-based and adjoint optimization algorithms. It exhibits strong robustness, global optimization capability, parallel computing potential, and a greater ability to escape local optima. Compared to the traditional NSGA-II algorithm, NSGA-IV expands the individual decision space, allowing for a wider range of choices and improving the diversity of the population for better inheritance. Details of the NSGA-IV algorithm can be found in reference [23].

To achieve global optimization of aerodynamic performance for both ROC and NOC operating conditions, this study combines Bernstein-based Bezier surface parameterization, the NSGA-IV algorithm, and CFD techniques. The optimized operating point is selected near the peak efficiency. The primary optimization objective is to maximize isentropic efficiency while

ensuring that the total pressure ratio remains above a specified constraint. Eqs (14) and (15) present the mathematical expressions for this objective and constraint. By integrating these methods, the optimization process effectively explores the design space, aiming to achieve the highest isentropic efficiency without compromising the total pressure ratio. This integrated approach leads to improved aerodynamic performance and enhanced efficiency of the centrifugal compressor blades.

Objective function:

$$\begin{cases} \max \eta_{ROC} \\ \max \eta_{NOC} \end{cases} \tag{14}$$

Constraints:

$$\begin{cases} \pi_{NOC\_opt} \geq 2.7 \\ \pi_{ROC\_opt} \geq 1.7 \\ x_i^L \leq x_i \leq x_i^U \end{cases} \tag{15}$$

where $\eta_{NOC}$ and $\eta_{ROC}$ are the isentropic efficiencies of the original impeller at NOC and ROC, respectively. $\pi_{NOC\_opt}$ is the total pressure ratio at the NOC after FFD optimization, and $\pi_{ROC\_opt}$ is the pressure ratio at the ROC after FFD optimization. $x_i$ is the optimization variable, $x_i^L$ the lower bound of the optimization variable, and $x_i^U$ is the upper bound of the optimization variable.

The Bernstein-based Bezier surface parameterization method employs two $6 \times 3$ order Bezier surfaces, one for each of the main blade and the manifold blade, to define their respective geometries. As shown in Fig 11, each surface has 7 control vertices in the $\xi$ direction (0, 0.1, 0.3, 0.5, 0.7, 0.9, 1.0) and 4 points in the $\eta$ direction (0, 0.4, 0.7, 1.0). To ensure the first-order continuity of the leading edge, fix $\xi_1$ and $\xi_2$ ($\xi_2'$) immobile. To ensure that the blade thickness remains consistent and avoids distortion, the change in control vertices for the

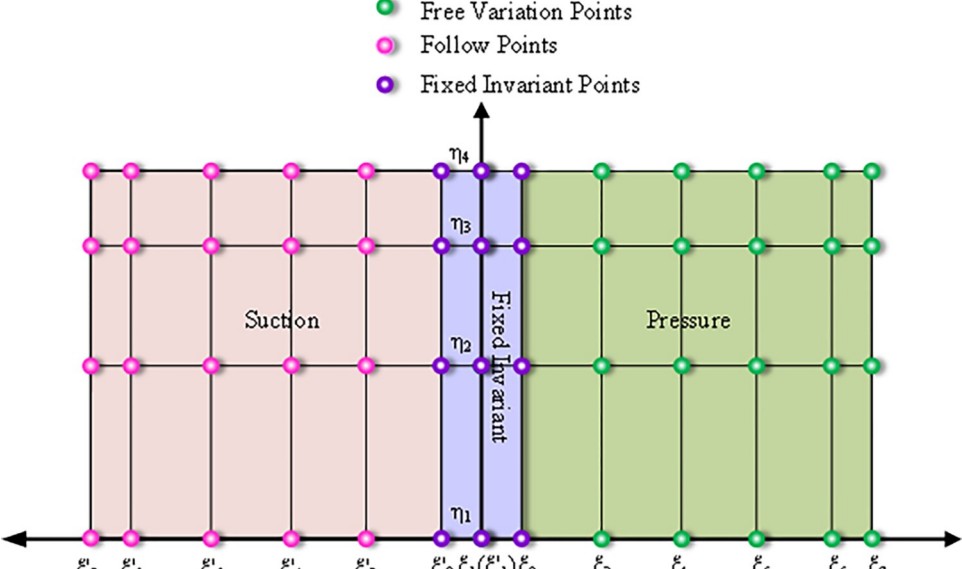

**Fig 11. Bezier surfaces control the distribution of vertices.**

**Table 2. NSGA-IV optimization algorithm parameter configuration.**

| Parameter | Value | Parameter | Value |
|---|---|---|---|
| Population size | 80 | Crossover Rate | 0.98 |
| Iteration number | 30 | Variation rate | 0.01 |

suction surface (pink) is synchronized with the corresponding change in control vertices for the pressure surface (green). The amount of change along the vertical plane of the green control points is used as the optimization variable. This approach results in a total of $(5 \times 4) \times 2 = 40$ optimization variables for both the main blade and splitter blade. The optimization algorithm parameters are defined in Table 2.

Fig 12 illustrates the global fast optimization process, which consists of the following steps:

1. Initialization: The initial population is generated using Latin hypercube design, ensuring good coverage of the design space.

2. Fitness Evaluation: Fitness values for the initial population are determined by applying the Bezier surface parameterization method and conducting CFD numerical calculations.

3. Offspring Generation: Cross-variance is performed to generate offspring individuals. The Bezier surface parameterization method and CFD numerical calculations are employed to obtain fitness values for the newly calculated offspring individuals.

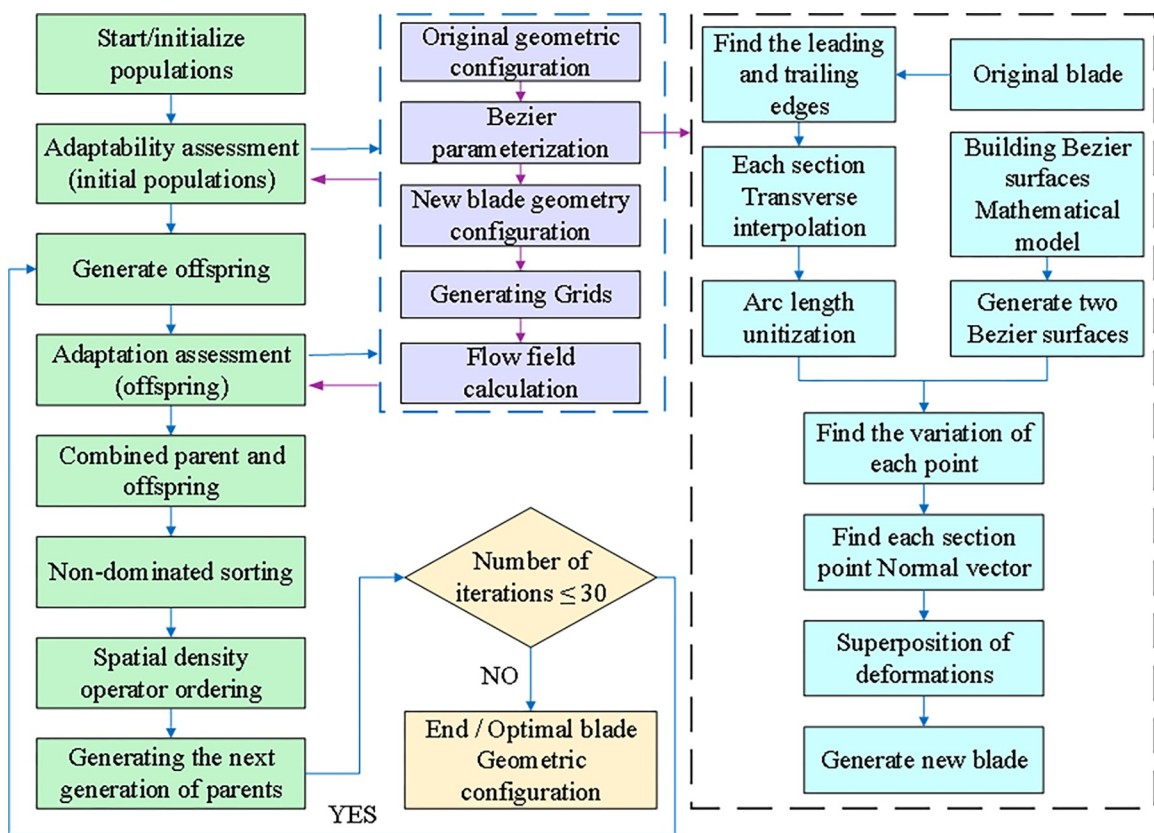

**Fig 12. Multi-condition global optimization process.**

4. Parent-Offspring Merging: The parent and offspring individuals are combined.

5. Parent Selection: Non-dominated sorting and spatial density operator sorting are applied to select the next generation of parent individuals.

6. Iteration: The process returns to the step 2 (Fitness Evaluation) and continues until the iteration requirements are met. The most suitable individual solution is selected based on the optimization objectives.

Through iterative implementation of these steps, the global fast optimization process effectively explores the design space and identifies the optimal solution. The use of Bezier surface parameterization and CFD numerical calculations ensures accurate fitness values. This approach enables the selection of optimized blade configurations with enhanced aerodynamic performance, leading to improved overall performance of the centrifugal compressor blades.

**First-stage global optimization results.**   Fig 13 illustrates the deformation cloud of the blade surface before and after global optimization. Examining the main blade along its span-wise direction reveals larger deformations at the tip and hub, with the middle section experiencing relatively minor changes. Along the chord length direction, the leading edge undergoes smaller changes. Specifically, the tip and middle section, towards the middle position, bend towards the suction surface, while the tip bends towards the pressure surface. At the trailing edge, the tip bends towards the suction surface, the middle section remains largely unchanged, and the hub bends towards the pressure surface.

The splitter blade exhibits similar deformation patterns along its span-wise direction, with larger changes at the tip and hub and minimal changes in the middle section. Along the chord length direction, the leading edge remains largely unchanged. At the middle position, the tip bends towards the suction surface, the middle section experiences minimal changes, and the hub bends towards the pressure surface. The trailing edge also shows similar patterns: the tip bends towards the suction surface, the middle section remains largely unchanged, and the hub bends towards the pressure surface.

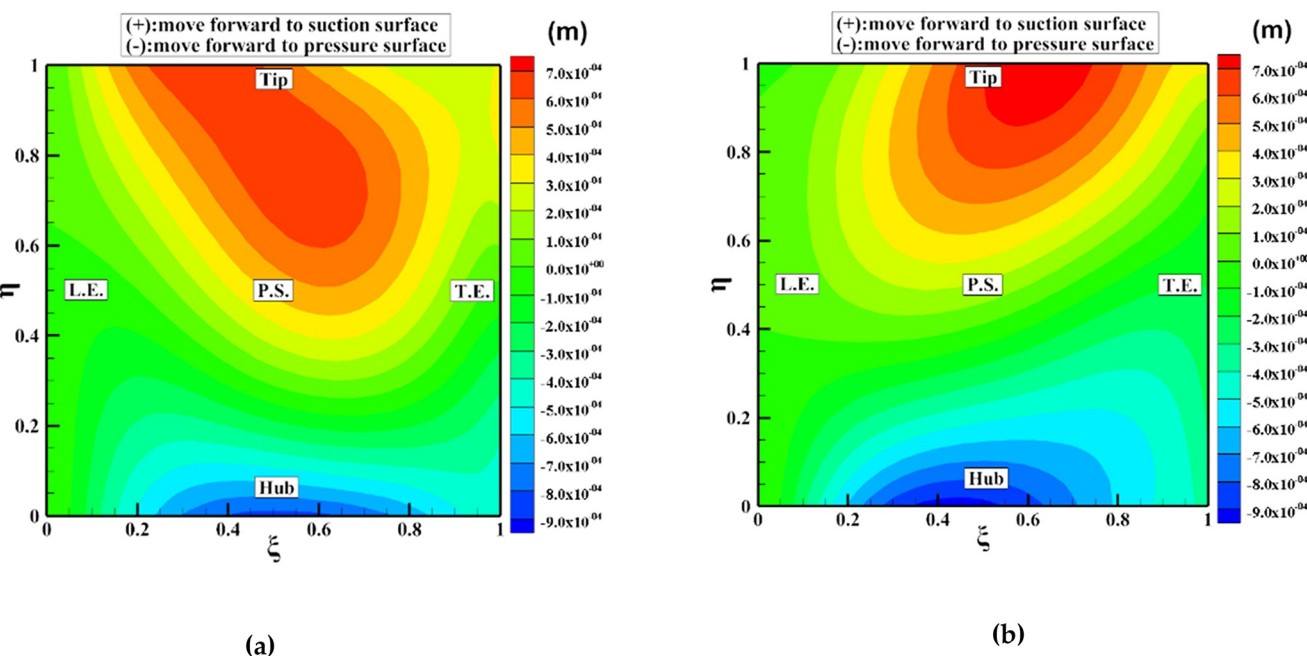

(a)                                                          (b)

**Fig 13.**  Deformation cloud plot: (a) Main blade, (b) Splitter blade.

**Table 3. Aerodynamic performance of ROC before and after the global optimization.**

|  | Mass flow (g/s) | Total pressure ratio | Isentropic efficiency |
|---|---|---|---|
| **Original** | 118.32 | 2.70 | 83.54% |
| **Optimized** | 128.76 | 2.72 | 84.83% |
| **Relative change** | +8.8% | 0.74% | +1.29% |

The optimized blade geometries and flow channel area modifications result in changes to aerodynamic performance and flow field structure. The optimized configurations exhibit altered characteristics designed to enhance overall aerodynamic performance and improve flow behavior. These changes aim to optimize flow patterns, reduce losses, and enhance flow efficiency through the centrifugal compressor blades. Ultimately, the optimized configurations contribute to improved aerodynamic performance, leading to increased efficiency, a larger surge margin, and overall enhanced compressor performance.

ROC: The comparison of aerodynamic performance before and after global optimization under multiple conditions is presented in Table 3. The results demonstrate significant improvements in various performance metrics. Specifically, the isentropic efficiency shows an enhancement of 1.29%, while the mass flow experiences an improvement of 8.8%. The total pressure ratio is enhanced by 0.74%, and the surge margin exhibits a notable increase of 6.2%. These improvements demonstrate the positive impact of the global optimization process on the aerodynamic performance of the centrifugal compressor blades.

NOC: The comparison of aerodynamic performance before and after global optimization under multiple conditions is presented in Table 4. The results reveal notable improvements in various performance parameters. The isentropic efficiency demonstrates an enhancement of 1.2%, while the mass flow experiences a significant improvement of 9.1%. The total pressure ratio shows a modest improvement of 0.24%, and the surge margin exhibits a substantial increase of 10%. Overall, the aerodynamic performance parameters of NOC demonstrate noticeable improvements as a result of the global optimization process conducted under multiple operating conditions.

Fig 14 compares performance curves before and after multi-case global optimization. The graphs show an overall upward shift in both isentropic efficiency and total pressure ratio curves for operating speeds of 70,000 rpm and 100,000 rpm. This upward shift indicates the success of the global optimization process, which focused on a reduced set of design variables in the first stage. The optimization has effectively improved the overall performance of the centrifugal compressor blades. The enhanced performance is evident across multiple operating conditions, further validating the effectiveness of the optimization process.

**Second-stage blade local optimization.** The B-spline based FFD 3D mesh parameterization offers the advantages of local strong support and flexible configuration, making it well-suited for the fine optimization of local geometric configurations in the second stage. This study combines the FFD parameterization method, the NSGA-IV algorithm [22], and CFD techniques to perform localized fine optimization of aerodynamic performance for both ROC and NOC.

**Table 4. Aerodynamic performance of NOC before and after the global optimization.**

|  | Mass Flow (g/s) | Total pressure ratio | Isentropic efficiency |
|---|---|---|---|
| **Original** | 77.36 | 1.7 | 85.53% |
| **Optimized** | 84.43 | 1.704 | 86.73% |
| **Relative change** | +9.1% | 0.24% | +1.2% |

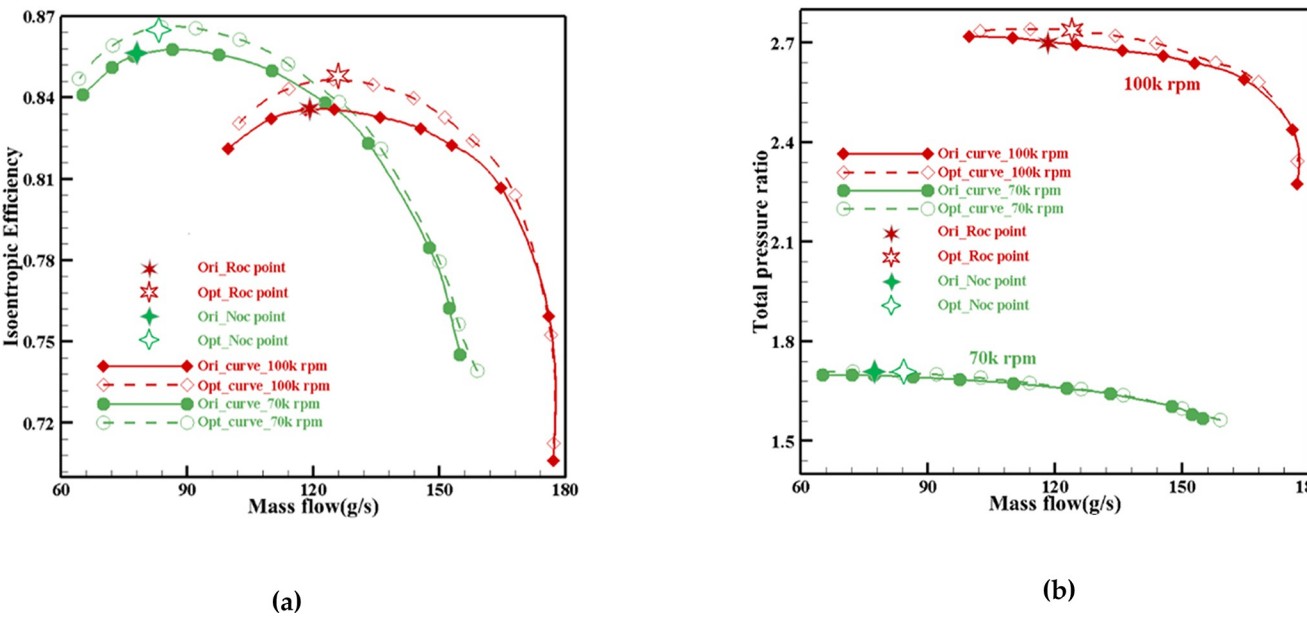

**(a)**                                                                                      **(b)**

**Fig 14.** Comparison of performance curves before and after global optimization for multiple operating conditions: (a) Flow Mass-Isentropic Efficiency, (b) Flow Mass-Total pressure ratio.

The second stage of optimization utilizes the impeller obtained after the first stage as the reference impeller. The optimization objective is to maximize the isentropic efficiency of the reference impeller simultaneously at ROC and NOC, while ensuring that the total pressure ratio remains equal to or higher than the original value. This constraint is expressed mathematically in Eqs (16) and (17).

By integrating the FFD parameterization method, the NSGA-IV algorithm, and CFD techniques, the local fine optimization process focuses on enhancing the aerodynamic performance of the centrifugal compressor blades while maintaining the desired total pressure ratio.

Objective function:

$$\begin{cases} \max \eta_{ROC\_Baseline} \\ \max \eta_{NOC\_Baseline} \end{cases} \tag{16}$$

Constraints:

$$\begin{cases} \pi_{NOC\_opt} \geq 2.7 \\ \pi_{ROC\_opt} \geq 1.7 \\ x_i^L \leq x_i \leq x_i^U \end{cases} \tag{17}$$

where $\eta_{NOC\_Baseline}$ is the NOC isentropic efficiency of the reference impeller at 70,000 rpm, $\eta_{ROC\_Baseline}$ is the ROC isentropic efficiency at 100,000 rpm. $\pi_{NOC\_opt}$ is the NOC total pressure ratio after FFD optimization, and $\pi_{ROC\_opt}$ is the ROC total pressure ratio after FFD optimization. $x_i$ is the optimized variable, $x_i^L$ is the lower limit of the optimized variable, and $x_i^U$ is the upper limit. The optimization process of the B-spline basis FFD parameterization method is shown in Fig 15, and the parameters of the optimization algorithm are configured as shown in Table 2.

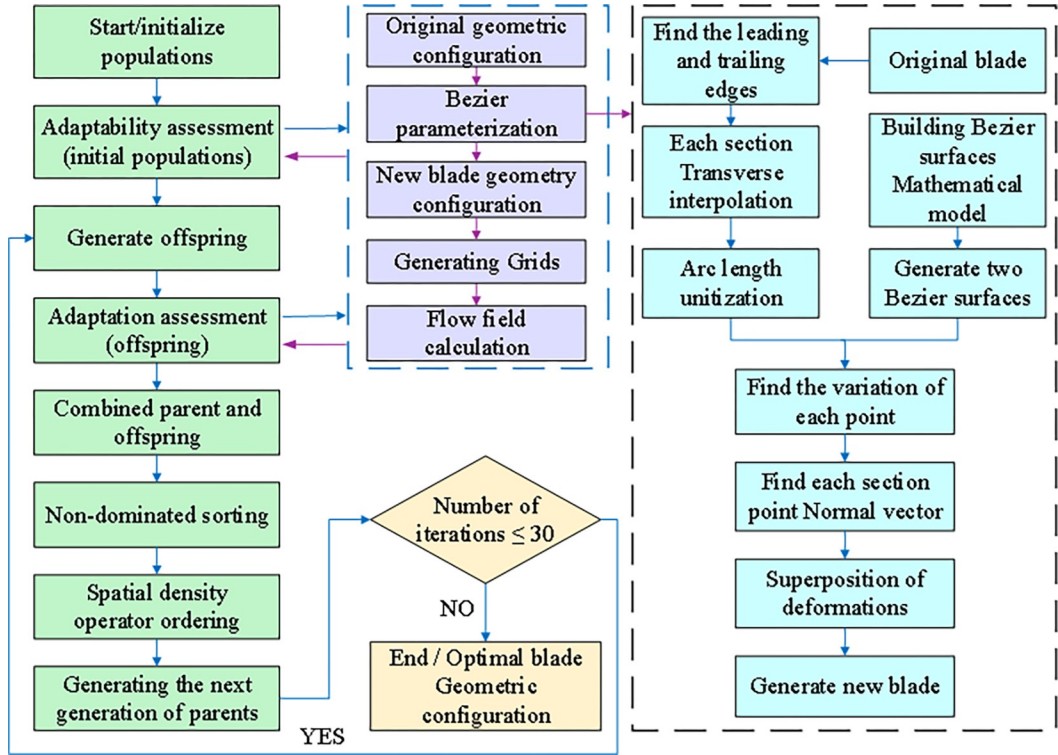

**Fig 15. Multi-condition local fine optimization process.**

(l) Separate control body frames are constructed for the main blade and the splitter blade. As shown in Fig 16, the control body frame for the main blade consists of 9 x 3 x 5 control points, while the control body frame for the splitter blade comprises 8 x 3 x 5 control points.

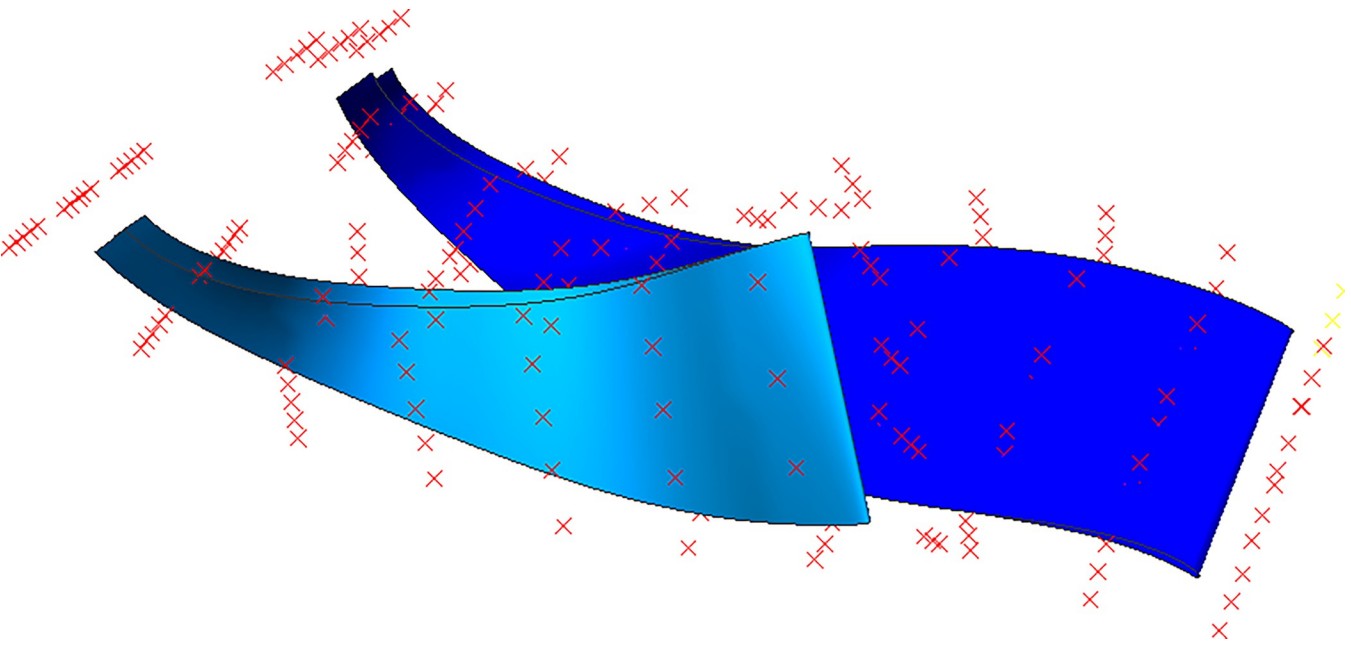

**Fig 16. FFD control body frame diagram.**

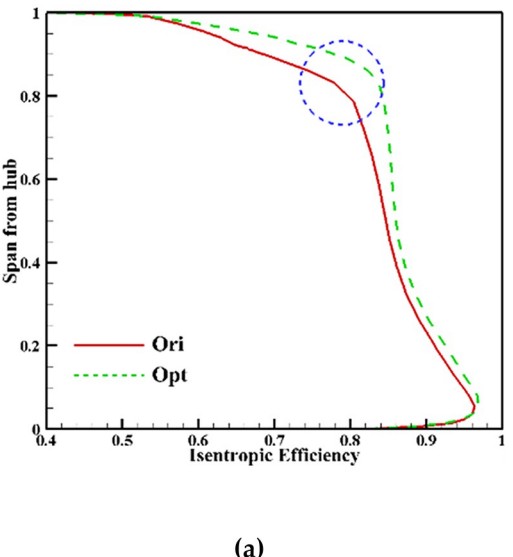
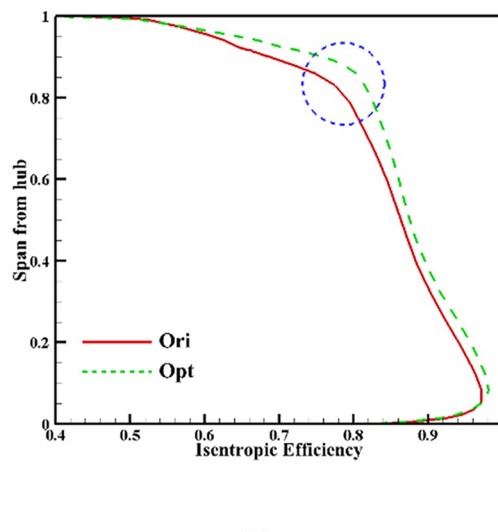

<div align="center">(a) (b)</div>

**Fig 17.** Isentropic efficiency distribution of the outlet: (a) ROC, (b) NOC.

(II) To target areas with significant optimization potential, specific control vertices (design vertices) are selected within a localized geometry region. This selection is based on an analysis of the baseline impeller's aerodynamic performance, focusing on identifying regions where improvements can be achieved.

Analysis of the study reveals a notable improvement at the blade tip of the original impeller after the first stage of global optimization. The improvement achieved is substantial, leaving limited room for further optimization, as depicted in Fig 17 (where "Ori" represents the original data and "Opt" represents the globally optimized data).

Based on conventional experience, it is well-known that the trailing edge of the centrifugal compressor blade has the most significant impact on aerodynamic performance, while the leading edge and middle sections also play a crucial role in influencing the flow field. Considering these factors, a total of 10 design points are selected for the trailing edge, while 8 design points are chosen for the leading edge and middle sections of the blade root. These design point positions (chordal, circumferential, radial) are detailed in Table 5 and visually represented by blue circle marks in Fig 18. This results in a total of 18 x 2 = 36 design variables.

To simplify computational processes and minimize the number of design variables, the search direction is defined by considering circumferentially adjacent control points, as illustrated in Fig 19. This selection reduces the degrees of freedom by two, effectively transforming the design variable dimension from three-dimensional to one-dimensional. This dimensionality reduction improves the efficiency of the search process. Furthermore, the exploration space

**Table 5. Position of the optimization variable.**

| Position | Position number (chordal, circumferential, radial) | | | | |
|---|---|---|---|---|---|
| **Leading Edge** | (1, 2, 1) | | | (1, 2, 2) | |
| **Middle part** | (6, 1, 1) | | (6, 2, 1) | (6, 3, 1) | |
| | (6, 1, 2) | | (6, 2, 2) | (6, 3, 2) | |
| **Trailing edge** | (9, 1, 1) | (9, 1, 2) | (9, 1, 3) | (9, 1, 4) | (9, 1, 5) |
| | (9, 3, 1) | (9, 3, 2) | (9, 3, 3) | (9, 3, 4) | (9, 3, 5) |

**(a)**

**(b)**

**(c)**

**(d)**

**Fig 18.** Optimization variable distribution chart: (a) Leading edge and middle design variables of the main blade, (b) Trailing edge design variables of the main blade, (c) Leading edge and middle design variables of the splitter blade, (b) Trailing edge design variables of the splitter blade.

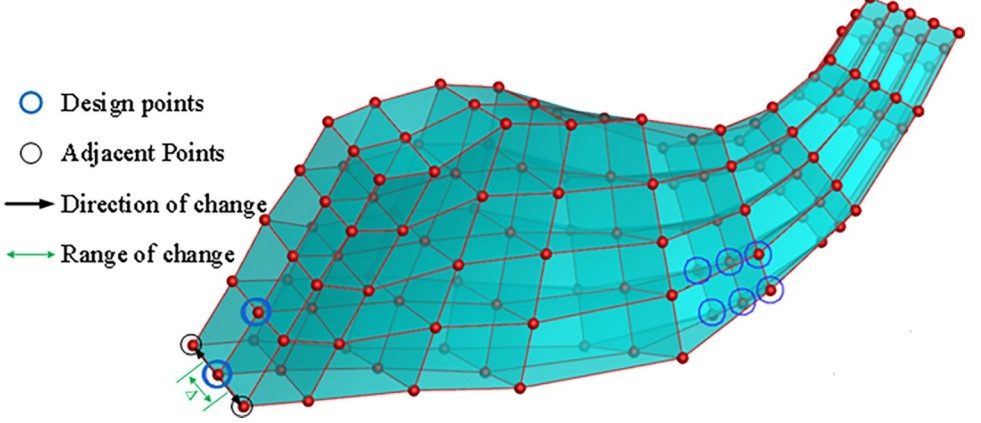

**Fig 19. Direction of change and range of change.**

is defined as half the distance between the design vertices and their circumferentially adjacent control points. This approach focuses exploration within a defined space, enabling more efficient optimization.

(III) The Latin hypercube design was employed to initialize the population of 80 individuals (parents). New impeller configurations were generated using the B-spline basis function FFD parameterization method. The fitness values of the parent individuals were then calculated by performing meshing and CFD simulations. This approach ensured a diverse initial population, covering a wide range of potential solutions and facilitating exploration of the optimization space.

(IV) The NSGA-IV multi-objective optimization algorithm is utilized to evolve the offspring population. The B-spline basis function FFD parameterization method is again employed to generate new impeller configurations. The fitness values of the individuals in the population are calculated by performing meshing and CFD simulations. This iterative optimization approach continuously improves impeller designs as the algorithm evolves and refines the population toward better-performing solutions.

(V) The parents and children populations are merged, and the selection process is performed to determine the best individuals. This selection is carried out using non-dominated sorting and spatial density operator sorting. The individuals that exhibit the best performance, considering both their dominance and spatial distribution, are selected as the parent individuals for the next iteration. The algorithm continues to iterate until the specified termination condition is met, ultimately leading to the identification of the best individual solution within the optimization process.

**Second-stage local optimization results.** Fig 20 illustrates the FFD mesh deformation diagram of the main blade before and after optimization. Notable deformations include: The control points at the leading edge of the leaf blade root bend towards the pressure surface, while the center control points bend towards the suction surface. At the root of the middle

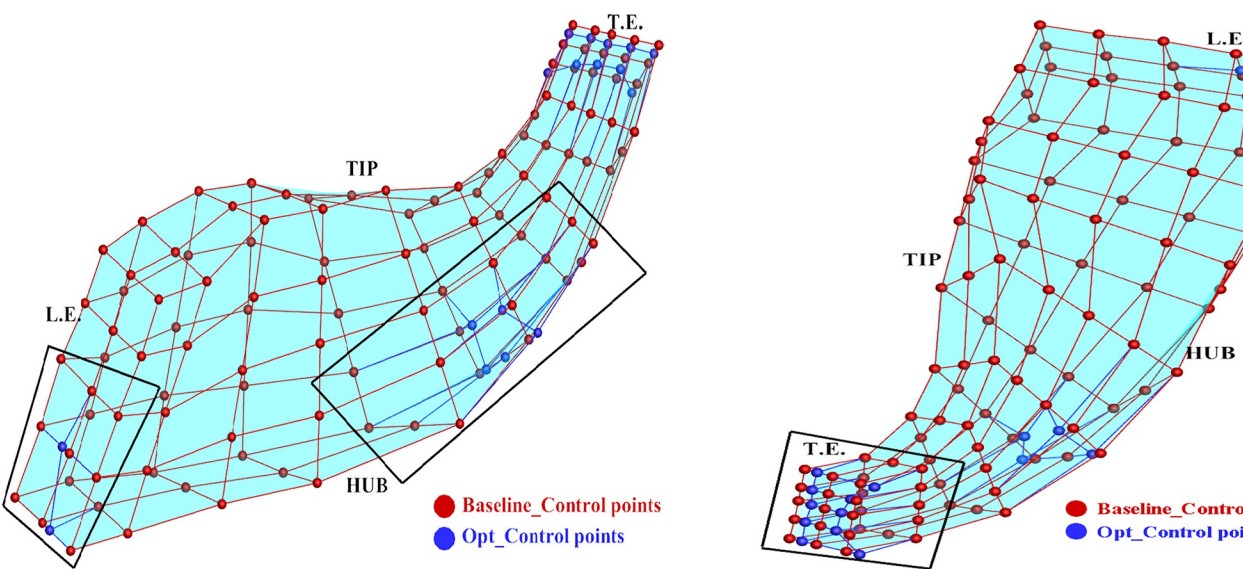

(a) Change of leading edge and middle control point of main blade

(b) Change of trailing edge control point of main blade

**Fig 20.** Changes in main blades before and after optimization: (a) Change of leading edge and middle control point of main blade, (b) Change of trailing edge control point of main blade.

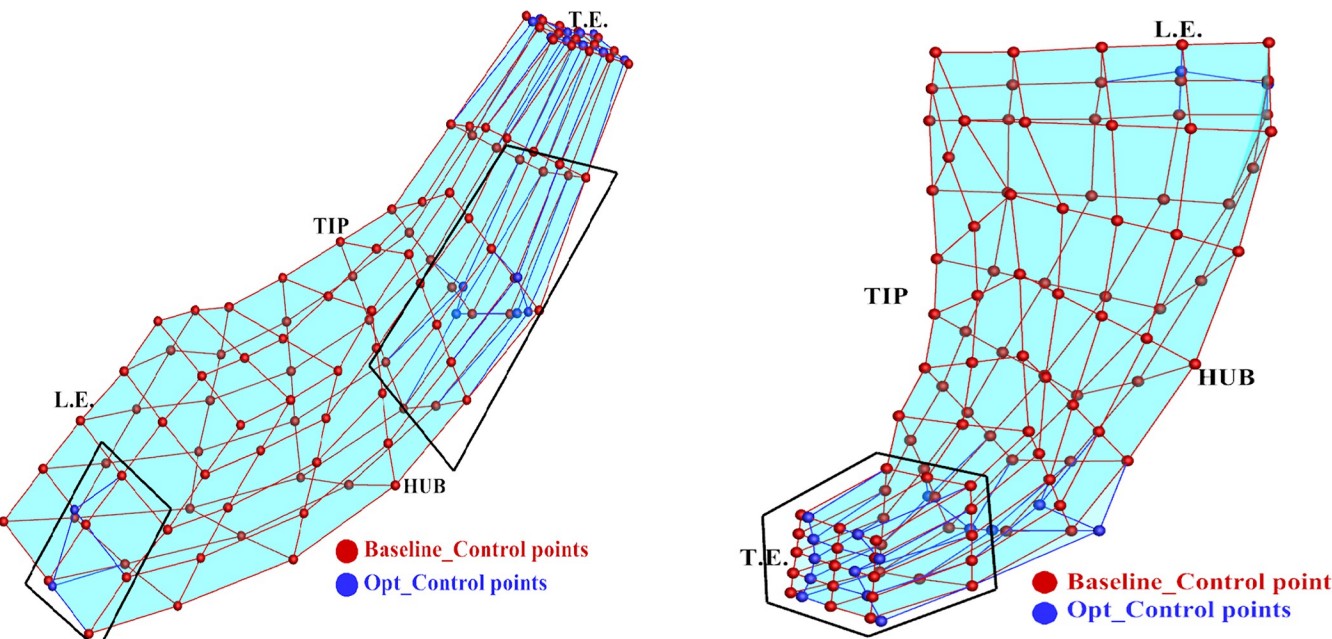

(a) Change of leading edge and middle control point of splitter blade

(b) Change of trailing edge control point of splitter blade

**Fig 21.** Changes in splitter blades before and after optimization: (a) Change of leading edge and middle control point of splitter blade, (b) Change of trailing edge control point of splitter blade.

blade, the distance between control points becomes smaller, resulting in a slight reduction in blade thickness. The trailing edge undergoes more significant changes, with the control points at the hub and tip of the blade expanding outward, causing an increase in blade thickness. Additionally, the distance between control points at the middle of the trailing edge becomes smaller, resulting in a slight reduction in blade thickness. These deformations highlight the modifications made to the blade geometry through the optimization process, which ultimately contribute to the desired improvements in aerodynamic performance.

Fig 21 illustrates the FFD mesh deformation of the splitter blade before and after optimization. Key changes observed include: The control points at the leading edge undergo minimal modifications. The control points at the central blade hub, as a whole, shift towards the suction surface. The control points at the blade hub of the trailing edge also shift towards the suction surface as a whole. Additionally, the distance between the control points at the middle of the trailing edge and the tip of the blade becomes smaller. The B-spline function FFD parameterization method ensures that only the specified area undergoes local deformation, while unspecified areas remain unchanged. This targeted deformation enables precise adjustments to the blade geometry, focusing on areas where aerodynamic performance improvements can be achieved.

Tables 6 and 7 present a comparison of the aerodynamic performance before and after optimization for both ROC and NOC, respectively. Fig 22 further illustrates this comparison by plotting the performance curves for the original impeller, globally optimized impeller, and locally optimized impeller. The isentropic efficiency curve for the locally optimized impeller shows an overall improvement compared to both the original and baseline impellers across multiple conditions. Similarly, the total pressure ratio curve exhibits slightly higher

**Table 6. ROC aerodynamic performance before and after optimization.**

| | Mass Flow(g/s) | Total pressure ratio | Isentropic efficiency | Surge margin |
|---|---|---|---|---|
| Original | 118.32 | 2.70 | 83.54% | 19.4% |
| Global optimization | 128.76 | 2.720 | 84.83% | 25.6% |
| Local optimization | 130.68 | 2.705 | 85.31% | 27.2% |
| Global increment | +8.8% | +0.74% | +1.29% | +6.2% |
| Local increment | +1.49% | -0.55% | +0.48% | +1.6% |
| Total incremental | +11.0% | +0.18% | +1.77% | +7.8% |

performance compared to the original impeller. These improvements demonstrate significant enhancements in aerodynamic performance achieved through both global and local optimization processes.

The flow field structure of NOC undergoes changes after local optimization, which can be observed as follows: In Fig 23, the cloud plot represents the entropy change at the downstream outlet S3 section before and after NOC optimization. The blue circles indicate areas of high entropy reduction after optimization, while the red circles represent areas of low entropy increase. Notably, the concentration of entropy improvement occurs between 0.1 times and 0.95 times of the blade height. This concentration is associated with various factors such as separation loss of the attached layer, secondary flow loss, and wake loss caused by the alterations in blade geometry, flow path, and back bend angle. These changes in the flow field structure reflect the optimization's impact on reducing losses and enhancing the overall performance of the NOC.

The flow field structure under NOC undergoes changes after local optimization, as illustrated in Fig 23. The cloud plot represents the entropy change at the downstream outlet S3 section before and after NOC optimization. The blue circles indicate areas of high entropy reduction after optimization, while the red circles represent areas of low entropy increase. A notable concentration of entropy improvement occurs between 0.1 and 0.95 times the blade height, potentially attributed to factors such as reduced separation losses from the attached layer, minimized secondary flow losses, and reduced wake losses resulting from modifications to the blade geometry, flow path, and back bend angle. These changes in the flow field structure reflect the optimization's impact on reducing losses and enhancing the overall performance of the impeller under NOC.

Fig 24 compares the isentropic efficiency distribution curves along the blade height direction at the downstream outlet before and after NOC optimization. The Figure reveals that the overall efficiency improves between 0.1x and 0.95x blade height after optimization, with the most significant improvement observed at 0.9x blade height. These results indicate that the optimization process effectively enhances isentropic efficiency throughout the blade height, particularly near the middle section.

**Table 7. NOC aerodynamic performance before and after optimization.**

| | Mass Flow(g/s) | Total pressure ratio | Isentropic efficiency | Surge margin |
|---|---|---|---|---|
| Original | 77.36 | 1.70 | 85.53% | 18.9% |
| Global optimization | 84.43 | 1.704 | 86.73% | 28.9% |
| Local optimization | 85.31 | 1.70 | 87.13% | 30.7% |
| Global increment | +9.1% | +0.24% | +1.2% | +10% |
| Local increment | +1.04% | -0.23% | +0.4% | +1.8% |
| Total incremental | +10.3% | +0.00% | +1.6% | +11.8% |

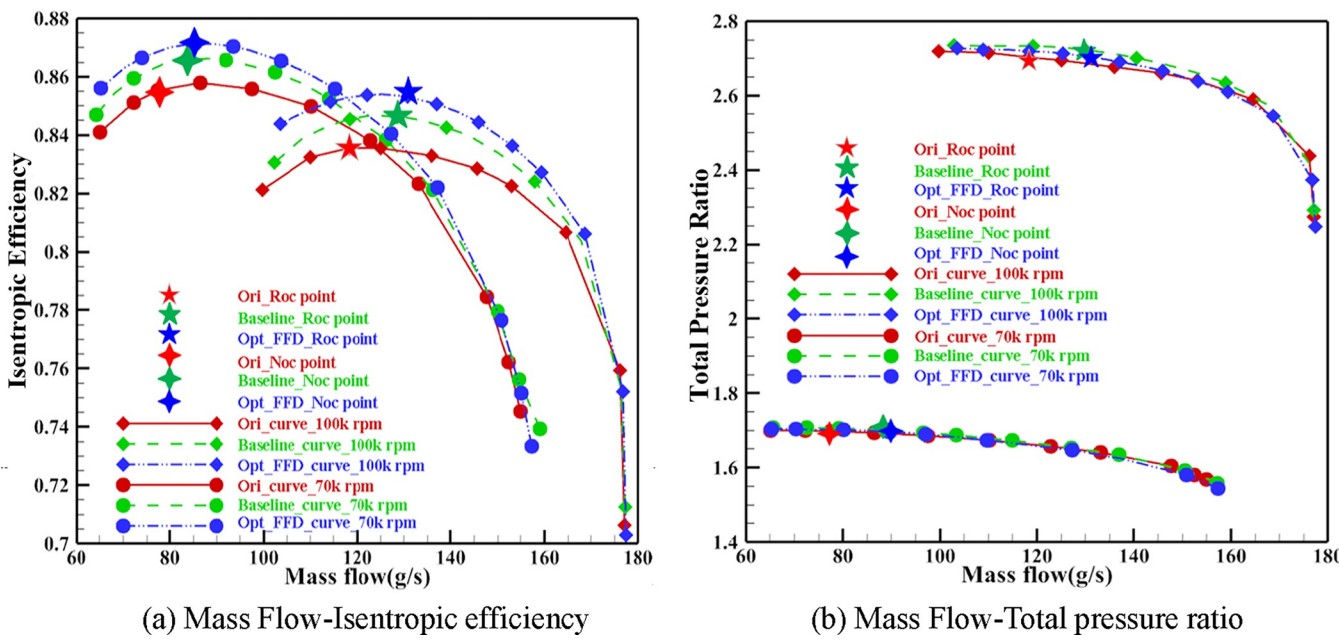

(a) Mass Flow-Isentropic efficiency  (b) Mass Flow-Total pressure ratio

**Fig 22.** ROC and NOC Performance Curve before and after optimization:(a) Mass Flow-Isentropic efficiency, (b) Mass Flow-Total pressure ratio.

Fig 25 compares the relative Mach number distribution at the 90% blade height B2B surface before and after local optimization under NOC. The results indicate the following changes: The inlet positive angle of attack matching exhibits a slight improvement. The maximum relative Mach number and the area of high Mach region are reduced, leading to a decrease in surge loss and an optimization of the downstream flow field. These changes are further supported by the increase in relative Mach numbers in the downstream exit regions E, F, and G, as well as the decrease in high entropy values at the tip, as shown in Fig 23. Overall, these

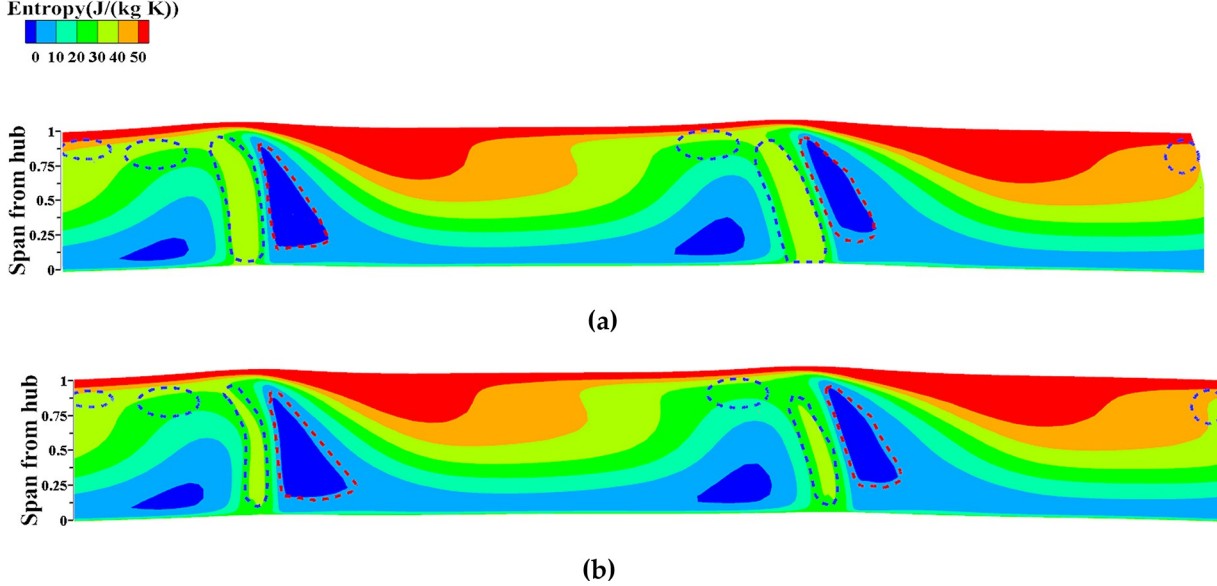

**Fig 23.** Entropy value of outlet S3 section after local optimization (NOC): (a) Original, (b) Optimized.

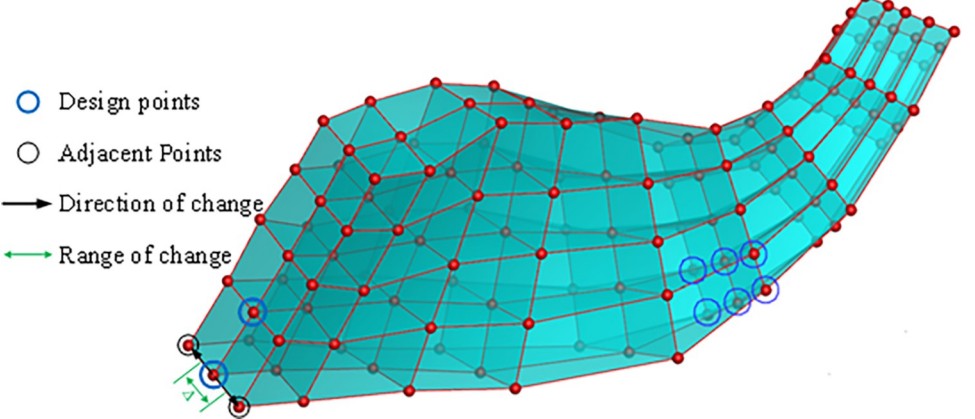

**Fig 24. Isentropic efficiency of outlet before and after local optimization (NOC).**

adjustments to the relative Mach number distribution contribute to an improved performance by reducing losses, shifting separation points, and optimizing the downstream flow field.

The flow field structure under ROC undergoes changes after local optimization. Fig 26 shows the entropy cloud diagram representing the changes in entropy at the S3 cross-section of the downstream outlet before and after optimization. Areas of entropy reduction are primarily concentrated in the hub and tip regions. To further analyze the impact of optimization on the flow field structure, a more detailed examination of the regions at 0.3x and 0.9x leaf height is conducted.

Fig 27 compares the relative Mach number distribution at 0.3 times the blade height under ROC before and after optimization. After optimization, the relative Mach number increases in the downstream region within the enclosed circle, indicating a reduction in separation losses and improved flow behaviour. Additionally, the flow channel between the splitter blade's

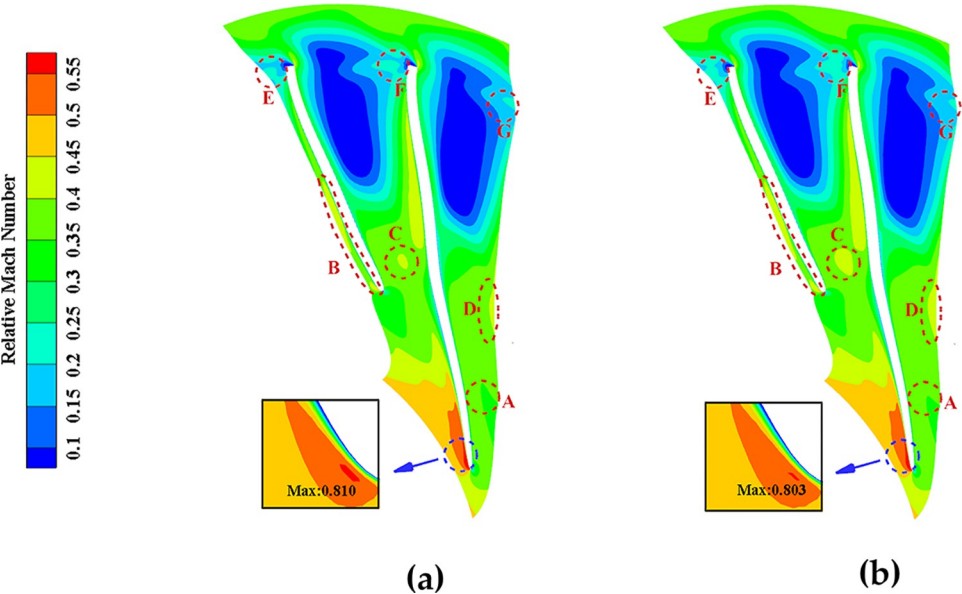

**Fig 25.** Relative Mach number of 90% height B2B surface before and after local optimization (NOC): (a) Original, (b) Optimized.

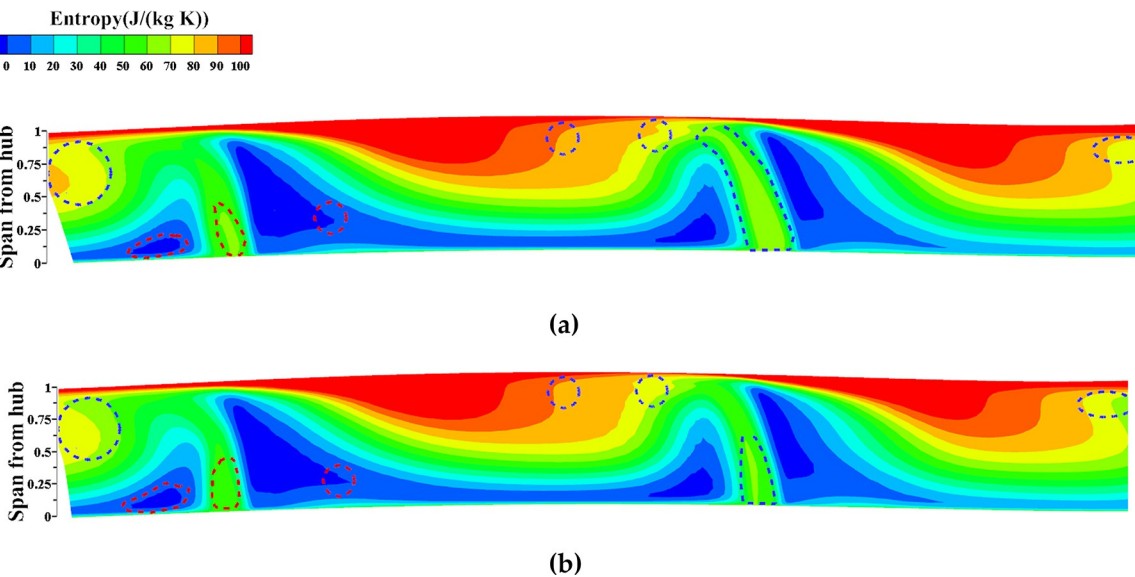

**Fig 26.** Entropy of downstream outlet S3 section after local optimization (ROC): (a) Original, (b) Optimized.

pressure surface and the main blade's suction surface narrows at the trailing edge, leading to a decrease in the adverse pressure gradient. These changes contribute to optimizing the flow field, resulting in improved performance by reducing losses and enhancing aerodynamic behaviour.

Fig 28 presents the entropy cloud at 0.9 times the blade height before and after local optimization (ROC). The optimization results indicate a significant reduction in the high entropy

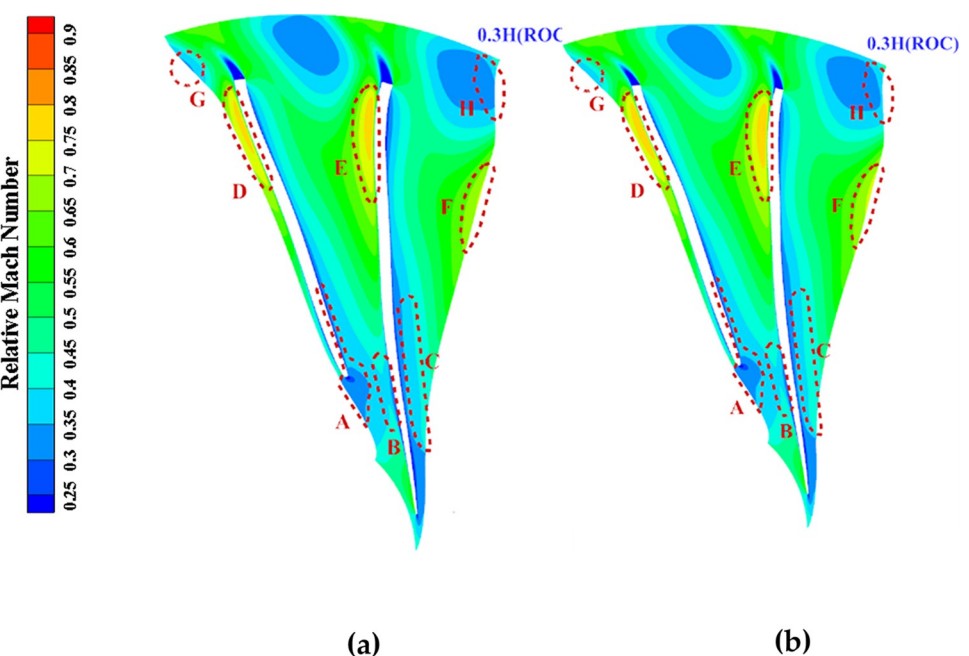

**Fig 27.** Relative Mach number of 30% height B2B surface before and after local optimization (ROC): (a) Original, (b) Optimized.

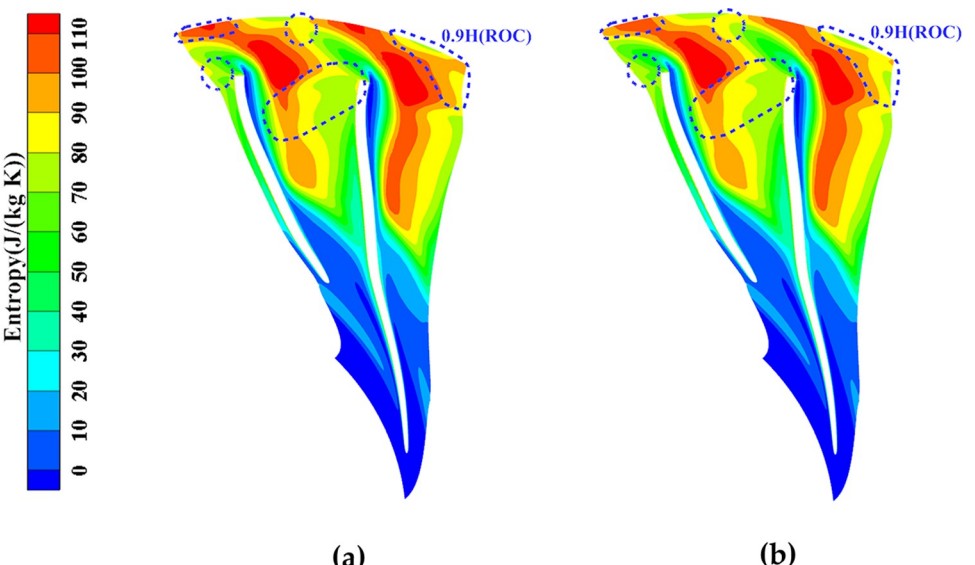

**Fig 28.** B2B entropy of 90% blade height before and after local optimization (ROC): (a) Original, (b) Optimized.

region. This reduction is in line with the slight thinning of the trailing edges of both the main blade and the splitter blade, which leads to a less expansive outlet section and a weakening of trailing losses. The improvement in the downstream outlet, as marked by the blue dashed box in Fig 28, further demonstrates the positive impact of local optimization on the flow field and overall aerodynamic performance.

## Conclusions

The refinement of centrifugal compressor blades through spline function parameterization, multi-objective evolutionary algorithms, and CFD simulations yielded the following conclusions:

The first stage of global optimization using the Bernstein-based Bezier surface parameterization method demonstrated significant improvements in aerodynamic performance. Under the rated operating condition (ROC), the isentropic efficiency increased by 1.2%, surge margin improved by 10%, mass flow improved by 9.1%, and total pressure ratio improved by 0.24%. Similarly, under the normal operating condition (NOC), the isentropic efficiency increased by 1.29%, surge margin improved by 6.2%, mass flow improved by 8.8%, and total pressure ratio improved by 0.74%. These improvements were validated by the upward shift in the overall aerodynamic performance curves, confirming the effectiveness of the global parameterization method.

The second stage of local optimization utilized the B-spline based FFD parameterization method, which offered flexibility, adaptability, and high-order continuity for fine-tuning the local blade geometry. The results further improved the aerodynamic performance, with a 0.48% increase in ROC isentropic efficiency, a 1.6% improvement in surge margin, and a 1.49% increase in mass flow rate. Similarly, for NOC, the isentropic efficiency improved by 0.4%, surge margin improved by 1.8%, and mass flow rate improved by 1.04%. These findings verified the effectiveness of the parameterization method for local optimization.

Compared to the original aerodynamic performance, the global and local optimization strategies yielded significant improvements. Under the constraints, the ROC isentropic efficiency improved by 1.77% and surge margin increased by 7.8%. For NOC, the isentropic

efficiency improved by 1.6%, and surge margin increased by 11.8%, demonstrating more pronounced improvements. By leveraging the advantages of Bezier and FFD spline functions, the global and local optimization approaches effectively exploited the blade's improvement potential with fewer design variables, resulting in a notable enhancement in solution quality.

Overall, the combination of spline function parameterization, multi-objective evolutionary algorithms, and CFD techniques proved to be a successful approach for the refined design optimization of centrifugal compressor blades, leading to significant improvements in aerodynamic performance and validating the effectiveness of the proposed methodologies.

## Acknowledgments

We sincerely thank all who contributed to this research. Special thanks to myself for his essential role in drafting and writing the paper. We appreciate Zixuan Sun for revising the article and providing valuable insights. Thanks to Jisheng Liu for his expertise in experimental design and execution. Our gratitude goes to Manxian Liu for his innovative ideas that enriched the study. Finally, we acknowledge Yong Zhou for his significant assistance with the experiments and writing.

## Author Contributions

**Data curation:** Yesong Wang, Jisheng Liu.

**Formal analysis:** Yesong Wang, Yong Zhou.

**Investigation:** Zixuan Sun, Manxian Liu.

**Methodology:** Yesong Wang, Jisheng Liu.

**Resources:** Yong Zhou.

**Software:** Yesong Wang, Zixuan Sun, Jisheng Liu.

**Validation:** Yesong Wang, Zixuan Sun.

**Writing – original draft:** Yesong Wang, Zixuan Sun, Jisheng Liu, Manxian Liu.

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
