## [Decision Letter · Decision Letter 0]

2 Jul 2024

PONE-D-24-17203Optimization Design of Centrifugal Impeller Based on Bezier surface and FFD Space Grid ParameterizationPLOS ONE Dear Dr. Zhou,

Thank you for submitting your manuscript to PLOS ONE. After careful consideration, we feel that it has merit but does not fully meet PLOS ONE’s publication criteria as it currently stands. Therefore, we invite you to submit a revised version of the manuscript that addresses the points raised during the review process.

We look forward to receiving your revised manuscript.

Kind regards,

Hasan Shahzad, Ph.D

Academic Editor

PLOS ONE

Journal Requirements:

2. We note that your Data Availability Statement is currently as follows: "All relevant data are within the manuscript and its Supporting Information files."

Reviewers' comments:

Reviewer's Responses to Questions

**Comments to the Author**

1. Is the manuscript technically sound, and do the data support the conclusions?

Reviewer #1: Partly

Reviewer #2: Yes

2. Has the statistical analysis been performed appropriately and rigorously? 

Reviewer #1: N/A

Reviewer #2: Yes

3. Have the authors made all data underlying the findings in their manuscript fully available?

Reviewer #1: Yes

Reviewer #2: Yes

4. Is the manuscript presented in an intelligible fashion and written in standard English?

Reviewer #1: Yes

Reviewer #2: No

5. Review Comments to the Author

Reviewer #1: The research paper titled "Optimization Design of Centrifugal Impeller Based on Bezier Surface and FFD Space Grid Parameterization" introduces a cutting-edge optimization design approach aimed at overcoming challenges associated with a high number of design variables, rigid configurations, and low optimization efficiency.

The utilization of a Bezier surface and Free-Form Deformation (FFD) model plays a crucial role in enhancing the design process. The FFD model was chosen for its ability to provide flexibility in manipulating the geometry of the impeller, allowing for efficient optimization and improved performance.

The introduction section of the paper can be enhanced by incorporating recent studies such as "CFD Analysis of a Vertical Axis Wind Turbine" (Mathematical Modelling of Fluid Dynamics and Nanofluids) to provide a solid grounding for the research context.

Comparative analysis with existing published works is essential to demonstrate the significance of the proposed methodology. By juxtaposing the results obtained in this study with those in the literature, a comprehensive evaluation can be made to showcase the advancements and contributions of the research.

The novelty of this work lies in the integration of Bezier surface and FFD model for optimizing the design of centrifugal impellers. This innovative approach offers a unique solution to the challenges faced in traditional design processes, leading to enhanced efficiency and performance.

In conclusion, the key findings of this research can be summarized into clear and impactful points highlighting the successful application of the proposed methodology, the improvements achieved in design efficiency, and the potential for broader implications in the field of impeller design. By emphasizing these key takeaways, the conclusion can effectively underscore the significance and implications of the study.

Reviewer #2: The over all it is a good study. But however, the presentation of the results should be improved. The quality of the graphs is not good. The language of article is also very weak it should be improved.

6. PLOS authors have the option to publish the peer review history of their article (what does this mean?). If published, this will include your full peer review and any attached files.

Reviewer #1: No

Reviewer #2: No

---

## [Author Response · Author response to Decision Letter 0]

24 Aug 2024

Responses to Journal 

（1）Comments：

https://journals.plos.org/plosone/s/file?id=wjVg/PLOSOne formatting_sample_main_body.pdf

Responses: 

Thank you for the comments.

We have ensured that our manuscript now fully adheres to PLOS ONE’s style requirements, including the correct file naming conventions. We have used the provided PLOS ONE style templates to format the main body, title page, and author affiliations appropriately.

（2）Comments：

We note that your Data Availability Statement is currently as follows: "All relevant data are within the manuscript and its Supporting Information files."

Please confirm at this time whether or not your submission contains all raw data required to replicate the results of your study. Authors must share the "minimal data set" for their submission. PLOS defines the minimal data set to consist of the data required to replicate all study findings reported in the article, as well as related metadata and methods (https://journals.plos.org/plosone/s/data-availability#loc-minimal-data-set-definition).

Authors do not need to submit their entire data set if only a portion of the data was used in the reported study. If your submission does not contain these data, please either upload them as Supporting Information files or deposit them to a stable, public repository and provide us with the relevant URLs, DOIs, or accession numbers. For a list of recommended repositories, please see https://journals.plos.org/plosone/ s/recommended-repositories. If there are ethical or legal restrictions on sharing a de-identified data set, please explain them in detail (e.g., data contain potentially sensitive information, data are owned by a third-party organization, etc.) and who has imposed them (e.g., an ethics committee). Please also provide contact information for a data access committee, ethics committee, or other institutional body to which data requests may be sent. If data are owned by a third party, please indicate how others may request data access.

Responses: 

Thank you for the comments.

We confirm that our submission contains all the raw data required to replicate the results of our study. These data are included within the manuscript and its Supporting Information files.

Responses to Reviewer #1

Comments：

The research paper titled "Optimization Design of Centrifugal Impeller Based on Bezier Surface and FFD Space Grid Parameterization" introduces a cutting-edge optimization design approach aimed at overcoming challenges associated with a high number of design variables, rigid configurations, and low optimization efficiency.

The utilization of a Bezier surface and Free-Form Deformation (FFD) model plays a crucial role in enhancing the design process. The FFD model was chosen for its ability to provide flexibility in manipulating the geometry of the impeller, allowing for efficient optimization and improved performance.

The introduction section of the paper can be enhanced by incorporating recent studies such as "CFD Analysis of a Vertical Axis Wind Turbine" (Mathematical Modelling of Fluid Dynamics and Nanofluids) to provide a solid grounding for the research context.

Comparative analysis with existing published works is essential to demonstrate the significance of the proposed methodology. By juxtaposing the results obtained in this study with those in the literature, a comprehensive evaluation can be made to showcase the advancements and contributions of the research.

The novelty of this work lies in the integration of Bezier surface and FFD model for optimizing the design of centrifugal impellers. This innovative approach offers a unique solution to the challenges faced in traditional design processes, leading to enhanced efficiency and performance.

In conclusion, the key findings of this research can be summarized into clear and impactful points highlighting the successful application of the proposed methodology, the improvements achieved in design efficiency, and the potential for broader implications in the field of impeller design. By emphasizing these key takeaways, the conclusion can effectively underscore the significance and implications of the study.

Responses: 

We truly appreciate this comment.

The proposed method combines the strengths of surface and spatial mesh parameterization to develop both the Bezier surface model and the free-form deformation (FFD) model. This approach enables flexible and precise reshaping of centrifugal compressor blades while significantly reducing the number of global design variables required for rapid optimization.

To strengthen the research background, we have incorporated several recent studies into the introduction and updated older references to those published within the last five years. Please refer to the text for signs of modification

This paper introduces, for the first time, the combined use of Bezier surface parameterization and FFD parameterization methods to optimize the design of centrifugal impellers. This approach can efficiently and rapidly enhance the overall aerodynamic performance of centrifugal pressurized impellers, offering significant improvements compared to using a single parameterization method.

Responses to Reviewer #2

Comments：

The over all it is a good study. But however, the presentation of the results should be improved. The quality of the graphs is not good. The language of article is also very weak it should be improved.

Responses: 

We truly appreciate this comment.

Thank you for your constructive feedback. We have addressed your concerns by improving the quality of the graphs to ensure they are clearer and more informative. Additionally, we have carefully reviewed and refined the language throughout the manuscript to enhance its readability and clarity. We believe these revisions significantly improve the presentation of our results and the overall quality of the paper. Please refer to the Revised Manuscript with Track Changes.

---

## [Decision Letter · Decision Letter 1]

8 Sep 2024

Optimization Design of Centrifugal Impeller Based on Bezier surface and FFD Space Grid Parameterization

PONE-D-24-17203R1

Dear Dr. Zhou,

We’re pleased to inform you that your manuscript has been judged scientifically suitable for publication and will be formally accepted for publication once it meets all outstanding technical requirements.

Kind regards,

Hasan Shahzad, Ph.D

Academic Editor

PLOS ONE

Additional Editor Comments (optional):

Reviewers' comments:

Reviewer's Responses to Questions

**Comments to the Author**

1. If the authors have adequately addressed your comments raised in a previous round of review and you feel that this manuscript is now acceptable for publication, you may indicate that here to bypass the “Comments to the Author” section, enter your conflict of interest statement in the “Confidential to Editor” section, and submit your "Accept" recommendation.

Reviewer #1: All comments have been addressed

2. Is the manuscript technically sound, and do the data support the conclusions?

Reviewer #1: Partly

3. Has the statistical analysis been performed appropriately and rigorously? 

Reviewer #1: Yes

4. Have the authors made all data underlying the findings in their manuscript fully available?

Reviewer #1: Yes

5. Is the manuscript presented in an intelligible fashion and written in standard English?

Reviewer #1: Yes

6. Review Comments to the Author

Reviewer #1: In general acceptable,

Now, I think that this manuscript can be accepted for publication in PLOS ONE

7. PLOS authors have the option to publish the peer review history of their article (what does this mean?). If published, this will include your full peer review and any attached files.

Reviewer #1: No

---

## [Editor Report · Acceptance letter]

12 Sep 2024

PONE-D-24-17203R1 

PLOS ONE

Dear Dr. Zhou, 

I'm pleased to inform you that your manuscript has been deemed suitable for publication in PLOS ONE. Congratulations! Your manuscript is now being handed over to our production team.

Kind regards, 

on behalf of

Dr. Hasan Shahzad 

Academic Editor

PLOS ONE